# Dependence of contextual modulation in macaque V1 on interlaminar signal flow

**Shude Zhu[1], Yu Jin Oh[1], Ethan B Trepka[1], Xiaomo Chen[1,2], Tirin Moore[1]\***

[1]Department of Neurobiology and Howard Hughes Medical Institute, Stanford University School of Medicine, Stanford, United States; [2]Center for Neuroscience, Department of Neurobiology, Physiology, and Behavior, Stanford, United States

## eLife Assessment

The results by Zhu et al provide **valuable** insights into the representation of border ownership in area V1. They used neuropixel recording to demonstrate the clustering of border ownership, and compared cross-correlation functions between neurons in different layers to demonstrate that they depend on the type of stimulus. The strength of the evidence is **solid** but can be improved by performing additional analyses and addressing some concerns (as raised in the previous and current review), and accounting for the differences in classical and non-classical receptive field stimulation conditions.

**\*For correspondence:**
tirin@stanford.edu

**Abstract** In visual cortex, neural correlates of subjective perception can be generated by modulation of activity from beyond the classical receptive field (CRF). In macaque V1, activity generated by nonclassical receptive field (nCRF) stimulation involves different intracortical circuitry than activity generated by CRF stimulation, suggesting that interactions between neurons across V1 layers differ under CRF and nCRF stimulus conditions. Using Neuropixels probes, we measured border ownership modulation within large, local populations of V1 neurons. We found that neurons in single columns preferred the same side of objects located outside of the CRF. In addition, we found that cross-correlations between pairs of neurons situated across feedback/horizontal and input layers differed between CRF and nCRF stimulation. Furthermore, independent of the comparison with CRF stimulation, we observed that the magnitude of border ownership modulation increased with the proportion of information flow from feedback/horizontal layers to input layers. These results demonstrate that the flow of signals between layers covaries with the degree to which neurons integrate information from beyond the CRF.

## Introduction

The classical receptive field (CRF) defines the region of the sensory periphery where appropriate stimulation evokes a spiking response in a neuron (*Hartline, 1938*; *Alonso and Chen, 2009*). Those evoked responses are largely driven by feedforward inputs from earlier stages of sensory processing. In macaque primary visual cortex (V1), neuronal activity is largely determined by ascending input from the dorsal lateral geniculate nucleus (dLGN) which arrives principally in layers 4Cα and 4Cβ (*Hubel and Wiesel, 1972*; *Hendrickson et al., 1978*; *Blasdel and Lund, 1983*; *Callaway, 1998*; *Callaway, 2004*). However, considerable evidence has established that the responses of sensory cortical neurons are also robustly influenced by stimulation concurrently presented outside of the CRF (*Allman et al., 1985*; *Fitzpatrick, 2000*; *Albright and Stoner, 2002*; *Roelfsema, 2006*; *Angelucci et al., 2017*). Specifically, whereas stimulation outside of the CRF alone fails to evoke spiking responses, such stimulation nonetheless alters the responses evoked by stimulation within the CRF. In the visual cortex,

these nonclassical receptive field (nCRF) effects often contribute to the neural correlates of visual perceptual phenomena, such as illusory contours (*Von der Heydt et al., 1984*; *Ramsden et al., 2001*), visual salience and boundary segmentation (*Knierim and van Essen, 1992*; *Sillito et al., 1995*; *Yan et al., 2018*; *Lee et al., 2002*; *Nothdurft et al., 1999*), contour integration (*Nelson and Frost, 1985*; *Kapadia et al., 1995*; *Li et al., 2006*), figure-ground segregation (*Lamme, 1995*; *Zipser et al., 1996*; *Poort et al., 2012*), and border ownership (*Zhou et al., 2000*; *von der Heydt, 2015*; *von der Heydt, 2023*).

In contrast to the feedforward mechanisms underlying the CRF, nCRF effects are thought to be generated either by feedback from neurons with larger CRFs in higher areas (*Hupé et al., 1998*; *Bullier et al., 2001*; *Bair et al., 2003*; *Nassi et al., 2013*; *Chen et al., 2017*; *Klink et al., 2017*; *Zhang et al., 2014*; *Nurminen et al., 2018*; *Keller et al., 2020*; *Pak et al., 2020*; *Vangeneugden et al., 2019*; *Gieselmann and Thiele, 2022*; *Chen et al., 2014*; *Van Kerkoerle et al., 2014*), or through intracortical horizontal connections (*Adesnik et al., 2012*; *Chisum et al., 2003*; *Stettler et al., 2002*), or by both (*Angelucci and Bressloff, 2006*; *Schwabe et al., 2006*; *Ichida et al., 2007*; *Angelucci et al., 2017*; *Liang et al., 2017*; *Self et al., 2013*). In contrast to feedforward inputs, feedback and horizontal inputs to V1 both avoid layer 4C and terminate predominantly in superficial (layers 1–3) and deep (layers 5–6) layers (*Rockland and Lund, 1983*; *Rockland and Pandya, 1979*; *Rockland and Virga, 1989*; *Anderson and Martin, 2009*; *Markov et al., 2014*; *Federer et al., 2021*; *Gilbert and Wiesel, 1983*; *Shmuel et al., 2005*; *Siu et al., 2021*). These differences between feedforward and feedback/horizontal circuitry suggest that the flow of signals across V1 layers should covary with the type of visual stimulation.

Newly developed, high-density Neuropixels probes have enabled recordings from a large, dense population of neurons (*Jun et al., 2017*; *Zhu et al., 2024*; *Trautmann et al., 2025*), and dramatically increase the quantity of identifiable functional interactions between pairs of neurons, particularly in non-human primates (*Trepka et al., 2022*). Recently, this approach was used to recapitulate known circuit properties, such as the pairwise lead-lag relationship between simple and complex cells and the canonical laminar input-output relationship in macaque V1 (*Trepka et al., 2022*), as well as the visual hierarchy in mouse visual cortical (*Siegle et al., 2021*; *Jia et al., 2022*) and subcortical areas (*Sibille et al., 2022*).

Here, using Neuropixels probes, we recorded the activity of hundreds of neurons simultaneously from single V1 columns in anesthetized macaques during CRF and nCRF stimulation. We leveraged the high yields obtained from different layers to examine the organization and circuitry underlying contextual modulation, specifically border ownership ($B_{own}$). $B_{own}$ is a form of contextual modulation in which neurons signal the occluding border between the background and a foreground object using stimulus information beyond the CRF. This function appears to be crucial for natural scene segmentation and object recognition (*Nakayama et al., 1995*). It has been shown that many neurons in early visual areas, including V1, respond differently to identical edges (borders) of objects when those objects lie at different locations outside the CRFs (*Zhou et al., 2000*; *Franken and Reynolds, 2021*; *Hesse and Tsao, 2016*; *Hesse and Tsao, 2023*). $B_{own}$ emerges rapidly after visual response onset (*Sugihara et al., 2011*), yet the relative contributions of feedforward, feedback, and horizontal circuits to $B_{own}$ remain unknown. Also unknown is whether $B_{own}$ is present under anesthesia. Several other types of contextual modulation have been demonstrated in anesthetized animals (*Ramsden et al., 2001*; *Hupé et al., 1998*; *Bair et al., 2003*; *Allman et al., 1985*; *Gilbert and Wiesel, 1990*; *Nothdurft et al., 1999*; *Webb et al., 2003*; *Zarella and Ts'o, 2016*; *Müller et al., 2003*; *Henry et al., 2013*; *Bijanzadeh et al., 2018*; although see *Lamme et al., 1998*) and can involve inter-cortical feedback (*Bijanzadeh et al., 2018*). Moreover, $B_{own}$ has been shown to occur independently of attention (*Qiu et al., 2007*; *O'Herron and von der Heydt, 2009*), and before object shape recognition (*Williford and von der Heydt, 2016*; *Ko and von der Heydt, 2018*), suggesting it may not rely on higher cognitive control or object recognition areas. Rather, $B_{own}$ may emerge through pre-attentive, automatic grouping mechanism in low/middle level visual areas, signifying a representation of 'proto-objects' in the early visual cortex (*Martin and von der Heydt, 2015*; *von der Heydt, 2023*; *Self et al., 2019*).

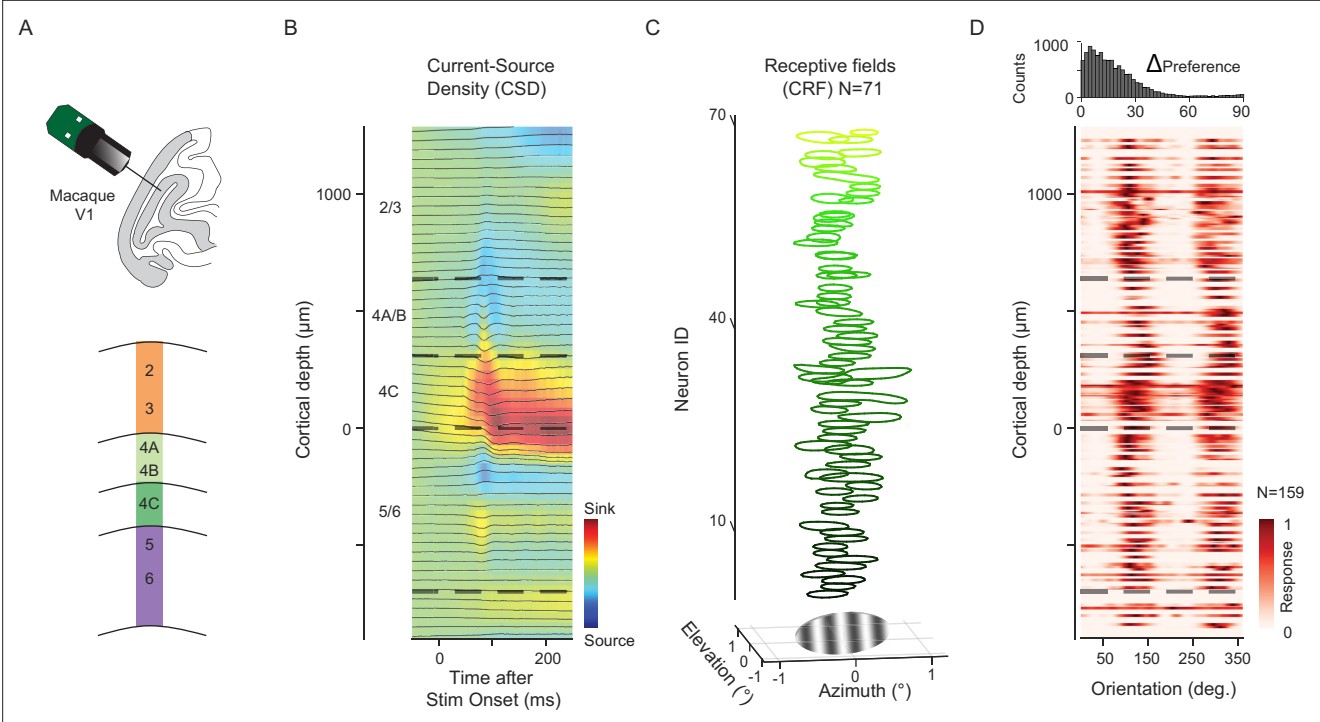

**Figure 1.** Classical receptive field (CRF) stimulation in single V1 columns. (**A**) Neuropixels probe (384 channels/3.84 mm) penetrations made into the lateral surface or underlying calcarine sulcus of macaque V1. Bottom: Diagram of designated laminar compartments. (**B**) Current-Source density (CSD) profile from an example recording session (M1, *pen1*), showing the early current sink (red) indicative of layer 4C. Laminar compartment boundaries (dashed lines) were determined using histological data and the CSD profile. (**C**) Top: 71 well-defined CRFs vertically stacked along the cortical depth, mapped with a 10x10 array (4x4 dva) of probe stimuli (gratings, 0.4 dva in diameter). Bottom: Drifting Gabor grating (1.5 dva in diameter) used for CRF stimulation, positioned within the joint receptive fields of recorded neurons. Zero (**x, y**) denotes the center of the CRF population. (**D**) Heatmap of visual responses from 159 neurons to 36 drift directions of Gabor gratings, vertically stacked along cortical depth (M1, *pen1*). Top: Distribution of differences in preferred orientation for all pairwise combinations of neurons.

The online version of this article includes the following figure supplement(s) for figure 1:

**Figure supplement 1.** Histological data for Neuropixels recordings in macaque V1.

# Results

## Classical RF stimulation in single V1 columns

Neuronal activity was recorded from two anesthetized rhesus macaques (M1 and M2, *Macaca mulatta*) using high-density, multi-contact Neuropixels probes (version 3A, IMEC Inc, Belgium; *Figure 1A*; Methods). These animals had no prior experience with awake experiments or exposure to the stimuli used in this study. Neuropixels probes were inserted into the lateral operculum of V1 with the aid of a surgical microscope at angles nearly perpendicular to the cortical surface. Prior to insertion, the probes were coated with a DiI derivative for subsequent histological visualization of the tracks (Methods; *Figure 1—figure supplement 1*). Four Neuropixels probe penetrations were made either in the opercular surface (M1, *pen 1–2*, ~4–6° eccentricities) or within the underlying calcarine sulcus (M2, *pen 3–4*, ~6–10° eccentricities) using the programmable channel selection feature of the Neuropixels probe. The boundaries of laminar compartments were estimated by combining current-source density (CSD) measurements and histological data (*Figure 1B*; *Figure 1—figure supplement 1*; Methods; *Zhu et al., 2024*; *Carr et al., 2025*). Each recorded neuron was assigned to one of four laminar compartments, specifically 5/6, 4C, 4A/B, 2/3 (mean thickness: 489, 281, 311, 650 μm, respectively, consistent with previous anatomical *O'Kusky and Colonnier, 1982*; *Lund, 1973* and CSD *Self et al., 2013*; *Chen et al., 2017* studies). We measured the responses from a total of 677 visually driven neurons (N=159, 181, 210, and 127 per recording) to CRF stimulation, and 621 neurons (N=142, 179, 152, and 148 per recording) to nCRF stimulation (Methods). As a result of the nearly perpendicular penetrations, the visual CRFs of V1 neurons were largely overlapping across the cortical depth (*Figure 1C*).

During each recording session, neurons were first tested with CRF stimuli consisting of drifting Gabor gratings with 1.5 degrees of visual angle (dva) in diameter, presented at the joint CRF location of simultaneously recorded neurons (Methods). In each of the four recordings, neurons across cortical depth exhibited very similar preferences to grating orientations (*Figure 1D*). The modal pairwise difference in preferred orientation between neurons across layers was near zero (*pen1*=5°; *pen2*=5°; *pen3*=1°; *pen4*=1°). This result confirmed that our recordings were nearly perpendicular and included neurons predominantly from single orientation columns. Gratings were sized to largely restrict the stimuli within the CRFs of recorded neurons to maximize the extent to which V1 responses were driven in large proportion by feedforward circuitry. In V1, geniculocortical inputs initially drive neurons within layers 4Cα and 4Cβ (*Hubel and Wiesel, 1972*; *Hendrickson et al., 1978*; *Blasdel and Lund, 1983*), which then propagate to supragranular and infragranular layers (*Yabuta and Callaway, 1998*; *Callaway, 2004*).

## Nonclassical RF stimulation and border ownership

Within each of the recordings, we also examined the responses of V1 neurons to nCRF stimuli. It is known that neurons in primate visual cortex integrate visual information from far beyond their CRFs (*Albright and Stoner, 2002*; *Allman et al., 1985*; *Fitzpatrick, 2000*; *Roelfsema, 2006*; *Angelucci et al., 2017*). One example of contextual modulation is the selectivity of neurons to border ownership ($B_{own}$). Specifically, neurons in early visual areas respond differently to identical edges (borders) of objects when those objects lie at different locations outside the CRFs (*Zhou et al., 2000*; *Franken and Reynolds, 2021*; *Hesse and Tsao, 2016*; *Hesse and Tsao, 2023*; *Figure 2*). As with other forms of nCRF modulation, $B_{own}$ is thought to emerge either by feedback from higher areas (*Craft et al., 2007*; *Jehee et al., 2007*; *Layton et al., 2012*; *Eguchi and Stringer, 2016*; *Wagatsuma et al., 2021*; *Mehrani and Tsotsos, 2021*) or from horizontal connections within the same area (*Zhaoping, 2005*; *Kogo et al., 2010*). In V1, feedback and horizontal inputs arrive principally within supragranular and infragranular layers, that is they avoid layer 4 (*Rockland and Pandya, 1979*; *Rockland and Lund, 1983*; *Rockland and Virga, 1989*; *Anderson and Martin, 2009*; *Markov et al., 2014*; *Federer et al., 2021*; *Gilbert and Wiesel, 1983*; *Shmuel et al., 2005*; *Siu et al., 2021*). Thus, V1 activity generated by nCRF stimulation is known to involve different circuits compared to the activity generated by CRF stimulation.

In this study, border ownership test stimuli consisted of uniformly white or black objects (squares, 8x8 dva) on a black or white background, respectively, as in previous studies (*Zhou et al., 2000*; Methods). One border of the object fell within the CRFs of the recorded neurons. Within the CRF, the oriented border could either be a dark-light edge or light-dark edge, resulting in two different local contrast polarity (LC) conditions (*Figure 2A*). Outside the CRF, the stimulus configuration differed for the same LC condition such that the oriented border could belong to opposite sides of an object relative to the CRF. Crucially, across these two object side conditions, the stimulus falling within the CRF was identical (*Figure 2B*) and was identical within an area that extended well beyond the CRF (8x16, dva). In this stimulus configuration, differences in evoked responses between the two object side conditions signify selectivity to border ownership (*Zhou et al., 2000*). The statistical significance was determined by a two-factor ANOVA on the mean spike counts for each condition, with object side (Side 1 and Side 2) and local contrast (LC1 and LC2) as the factors, along with their interaction (Methods).

We observed neurons with $B_{own}$ in our recordings from macaque V1. *Figure 2C and D* shows examples of four neurons selective to $B_{own}$. Each of these neurons exhibited a preference for LC, with two neurons preferring one contrast polarity and two preferring the other. In addition, each neuron responded differently depending on which side of the object appeared relative to the CRF, thus exhibiting $B_{own}$. Three of the neurons responded more vigorously to borders belonging to the lower-left side of object (Side 1; $p<10^{-14}$, $10^{-10}$, $10^{-3}$ for neurons 1, 2, and 4, respectively), while one neuron responded more vigorously to the border belonging to the upper-right side of the object (Side 2; $p=0.014$, neuron 3). Furthermore, one of the neurons showed a consistent preference for Side 1 across both LCs (neuron 4).

To quantify the $B_{own}$, as in previous studies (*Zhou et al., 2000*), we calculated a $B_{own}$ modulation index, defined as the difference in neuronal responses to borders belonging to opposite sides of objects, normalized by the maximum response for each neuron (Methods). Responses to the same side

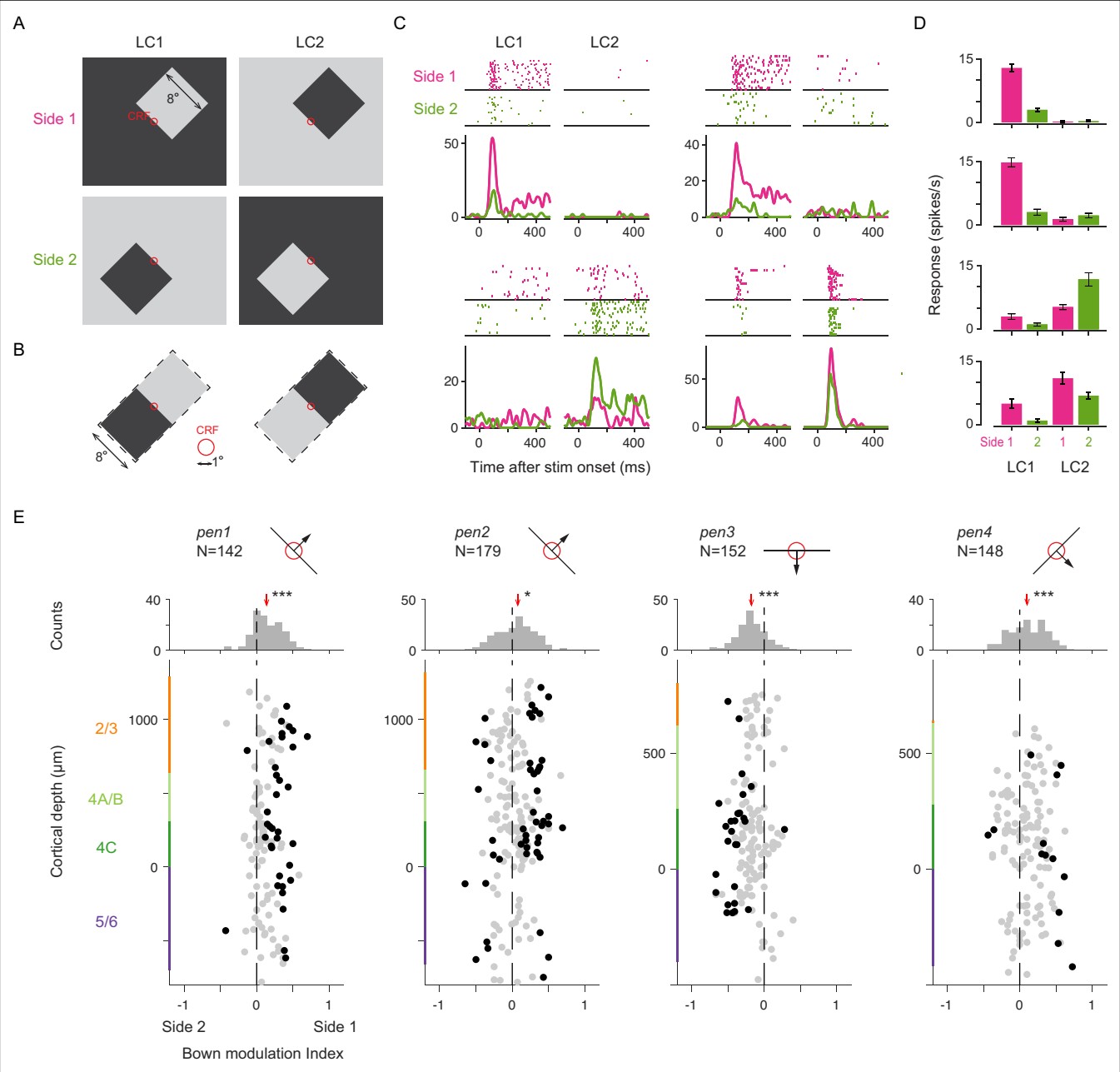

**Figure 2.** Nonclassical receptive field (nCRF) stimulation and border ownership. (**A**) Border ownership test stimuli for nCRF stimulation. Stimuli consisted of uniformly white or black objects (8° squares) on black or white backgrounds, respectively. One border of the object fell within the CRFs (red circle) of the recorded neurons, with orientation varying in 45° steps. Within the CRF, depending on local contrast polarity (LC), the border could either be a dark-light edge (LC1, Left column) or a light-dark edge (LC2, Right column). For the same LC condition, the border within the CRF could be the bottom-left edge (Side 1, top row) or top-right edge (Side 2, bottom row) of the object. (**B**) Enlarged view of stimulus configuration within the CRF, showing identical stimulus regions between Side 1 and Side 2 for each LC condition (8x16 dva). (**C**) Four example $B_{own}$ neurons recorded from V1. For each neuron, dot raster plots (top) and instantaneous firing rates (bottom) show responses to the four stimulus conditions depicted in (**A**), respectively. Magenta represents Side 1; green represents Side 2. (**D**) Bar plots showing the mean spike rates of the four neurons across the four stimulus conditions. Error bars denote SEM. (**E**) Top: Cartoon depicting the preferred border orientation for each of the four recording sessions (*pen*1-4). Arrows indicate the preferred side of objects relative to the border in the CRF. Bottom: $B_{own}$ modulation index for each neuron plotted at its recorded cortical depth across the recordings. Color bars on the ordinate represent laminar compartments. Black dots denote statistically significant $B_{own}$ neurons, determined by ANOVA with object side and local contrast as factors (p<0.05). Middle: Marginal distribution of the $B_{own}$ modulation index, red arrows indicating the population median. *p<0.05; ***p<$10^{-5}$.

of the object, but under opposite LC conditions, were averaged together to control for global luminance differences in the overall display. Across the 621 neurons recorded in four sessions, we found that 133 neurons (21.4% overall; *pen1*=28.2%; *pen2*=27.9%; *pen3*=18.4%; *pen4*=8.4%) responded differently to identical borders yet belonging to opposite sides of the object, a proportion consistent with previous studies in awake monkeys (*Zhou et al., 2000*). Neurons with $B_{own}$ were found within all cortical laminar compartments, including input layer 4 C (5/6: 28/144=19.4%; 4 C: 48/212=22.6%; 4 A/B: 19/114=16.7%; 2/3: 28/99=28.3%; *Supplementary file 1*), and the proportion was independent of laminar compartment ($\chi^2$(3)=4.78, p=0.19, chi-square test of homogeneity). Of all V1 neurons, 96 (15.5%) exhibited $B_{own}$ depending on the LC (5/6: 11.1%; 4 C: 17.9%; 4 A/B: 14.0%; 2/3: 20.2%), while 37 (6.0%) showed significant effects independent of local contrast (5/6: 8.3%; 4 C: 4.7%; 4 A/B: 2.6%; 2/3: 8.1%). Thus, the majority of V1 neurons with $B_{own}$ exhibited selectivity for only one local contrast, and neurons with $B_{own}$ invariant to local contrast were rare, also consistent with previous observations (*Zhou et al., 2000*).

## Columnar organization of border ownership selectivity

We next considered whether neurons with $B_{own}$ comprised a distinct group, or instead reflected a general tendency of neurons to prefer one side of the object, with statistically significant neurons falling at the tails of the distribution. We leveraged the large number of simultaneously recorded single neurons to measure the distribution of $B_{own}$ modulation across laminar compartments in each session. We found that in each of the four recordings, the distributions of $B_{own}$ modulation generally appeared unimodal, with neurons exhibiting $B_{own}$ falling at the distribution tails (*Figure 2E*). Surprisingly, we also found that within individual recordings, neurons across the cortical depth tended to share similar preferences for the side of the object. That is, for each recording, the $B_{own}$ modulation index differed significantly from 0 (Median modulation index: *pen1*=0.14, p<10$^{-12}$; *pen2*=0.08, p=0.04; *pen3*=−0.18, p<10$^{-15}$; *pen4*=0.10, p<10$^{-5}$, sign test against zero-median). For example, the first recording (*pen1*) was performed in a column in which most neurons preferred an orientation of 135°. In addition, within that column, neurons tended to exhibit higher responses to a border when it belonged to the lower-left side of an object (side 1), as opposed to the upper-right side (side 2). Thus, V1 exhibited a continuum of $B_{own}$, and neurons within single columns tended to prefer the same side of objects located outside of the CRF.

## Interlaminar cross-correlations among neurons within single V1 columns

The large number of simultaneously recorded neurons from each session also allowed us to measure interlaminar information flow among single neurons under different visual stimulation conditions. Temporally precise cross-correlations in spike trains offer a unique means of assessing functional interactions among neurons in neural circuits (*Perkel et al., 1967*). Cross-correlations can be interpreted as suggesting one of myriad putative circuit arrangements among neuronal ensembles (*Moore et al., 1970*; *Aertsen and Gerstein, 1985*; *Melssen and Epping, 1987*; *Ostojic et al., 2009*). The identification of such interactions has played an important role in elucidating neural circuits in the mammalian visual system (*Ts'o et al., 1986*; *Schwarz and Bolz, 1991*; *Clay Reid and Alonso, 1995*; *Alonso et al., 1996*; *Alonso and Martinez, 1998*; *Usrey et al., 1998*; *Nelson et al., 1992*; *Briggs et al., 2013*; *Roe and Ts'o, 1999*; *Senzai et al., 2019*; *Denman and Contreras, 2014*; *Siegle et al., 2021*; *Jia et al., 2022*; *Trepka et al., 2022*; *Smith and Kohn, 2008*; *Sibille et al., 2022*). In particular, it has been useful in specifying the flow of signals through local neural circuits (*Trepka et al., 2022*) and distributed networks (*Siegle et al., 2021*; *Jia et al., 2022*; *Sibille et al., 2022*). Notably, past studies have demonstrated that neuronal cross-correlations are dynamic and can depend on stimulus context and behavioral variables (*Hembrook-Short et al., 2019*; Martin and *Martin and von der Heydt, 2015*; *Hung et al., 2007*; *Hirabayashi and Miyashita, 2005*; *Hirabayashi et al., 2010*; *Steinmetz et al., 2000*).

Given that V1 activity generated by nCRF stimulation is thought to involve different circuitry than activity generated by CRF stimulation (*Angelucci et al., 2017*), we next asked if the pattern of cross-correlations measured under classical and nonclassical stimulus conditions might differ in a manner consistent with models of their underlying circuitry. Specifically, modulation of visual responses during nCRF stimulation, such as border ownership stimuli, is thought to emerge from either feedback from higher areas or horizontal connections within the same areas (*Bair et al., 2003*; *Bullier et al., 2001*;

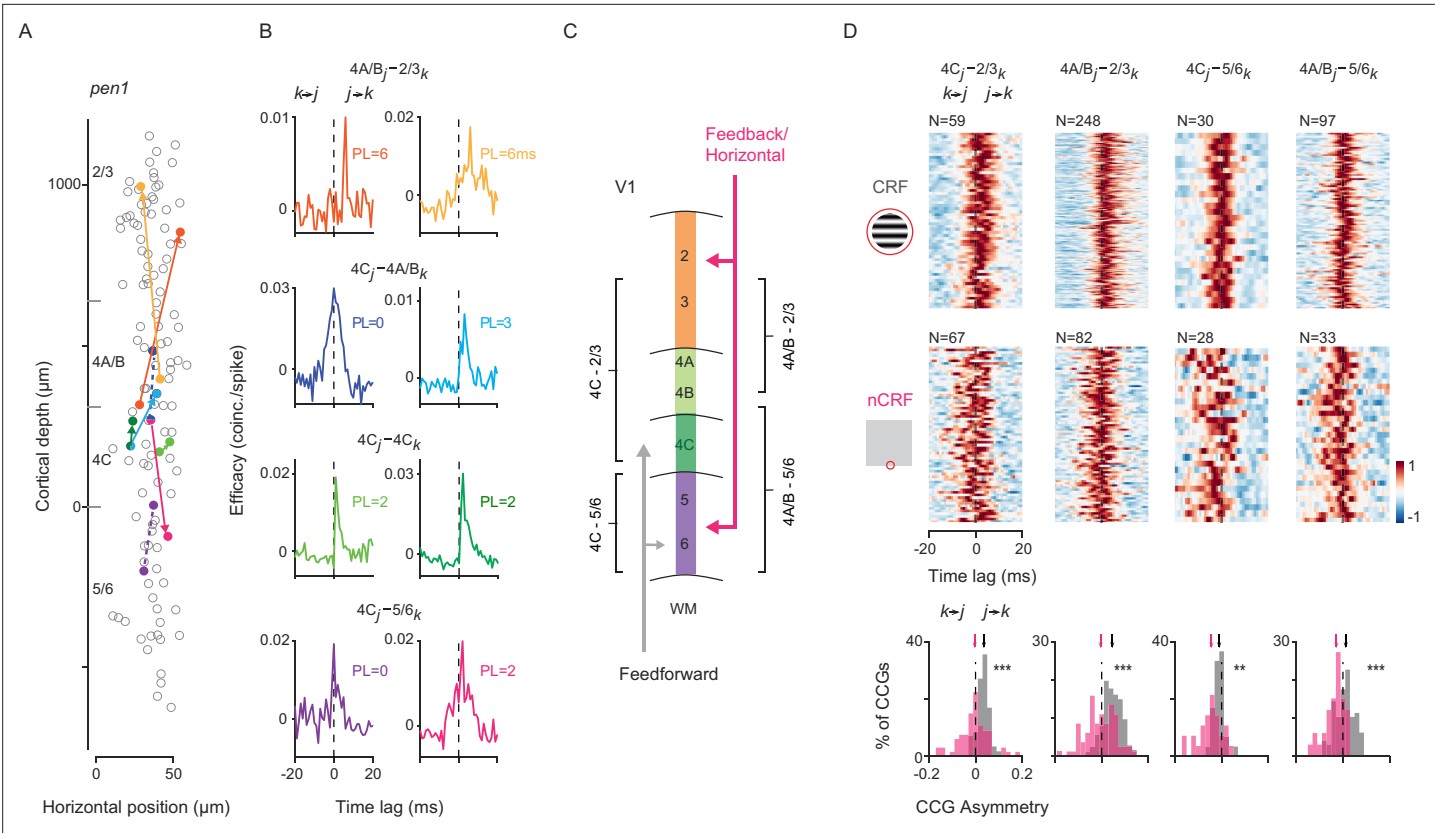

**Figure 3.** Interlaminar cross-correlations during CRF and nCRF stimulation. (**A**) Example recording (M1, *pen1*) showing 159 visually responsive neurons (circles) recorded simultaneously during CRF stimulation, plotted at their relative cortical depth. The abscissa is magnified for visualization. Also shown are eight example neuronal pairs with significant CCGs during CRF stimulation. Neuronal pairs with zero-time lag CCGs are connected by dashed lines, while pairs with nonzero-time lag CCGs are connected by arrowed lines, with the arrow pointing to the lagging neuron. (**B**) Eight corresponding CCGs from (**A**). Neuronal pairs are separated by their laminar compartment combinations. The reference neuron (1st neuron in the CCG function) is designated as $j$, and the target neuron as $k$. PL (peak lag, ms) is defined as the relative time delay in peak correlation between the two spike trains. (**C**) Diagram illustrating feedforward and feedback (and horizontal) projections within the designated laminar compartments. Feedforward inputs arrive within laminar compartments 4C, while feedback (and horizontal) projections principally target superficial and deep layers. (**D**) Heatmap of all significant CCGs during either CRF stimulation (Gabor gratings, top row) or nCRF stimulation (border ownership, middle row), for the four key interlaminar combinations (columns), calculated from one example recording (M1, *pen1*). Individual CCGs are normalized by their absolute maximum for visualization, and different CCGs are vertically stacked. Bottom row: Distribution of CCG asymmetry for each of the four key interlaminar compartments as in (**A**), across CRF stimulation (gray) and nCRF stimulation (magenta). Arrows indicate the population median. **p<0.002; ***p<10⁻⁵.

Hupé et al., 1998; Chen et al., 2017; Gieselmann and Thiele, 2022; Keller et al., 2020; Klink et al., 2017; Nassi et al., 2013; Bijanzadeh et al., 2018; Angelucci et al., 2017; Schwabe et al., 2006; Ichida et al., 2007; Schwabe et al., 2006; Pak et al., 2020; Vangeneugden et al., 2019; Zhang et al., 2014; Adesnik et al., 2012; Chisum et al., 2003; Stettler et al., 2002; Liang et al., 2017; Craft et al., 2007; Jehee et al., 2007; Chen et al., 2014; Self et al., 2013; Van Kerkoerle et al., 2014), both arriving principally in superficial (layers 1–3) and deep (layers 5–6) layers (Rockland and Lund, 1983; Rockland and Pandya, 1979; Rockland and Virga, 1989; Markov et al., 2014; Federer et al., 2021; Anderson and Martin, 2009; Gilbert and Wiesel, 1983; Shmuel et al., 2005; Siu et al., 2021). In contrast, CRF responses are driven predominantly by feedforward circuitry and geniculocortical inputs to layers 4C (Hubel and Wiesel, 1972; Hendrickson et al., 1978; Blasdel and Lund, 1983; Callaway, 1998; Callaway, 2004). This suggests that the pattern of cross-correlations among neurons across V1 layers may differ under CRF and nCRF stimulus conditions.

Across recording sessions, we calculated cross-correlograms (CCGs) from all possible pairwise combinations of neuronal spike trains during both CRF (Gabor gratings) and nCRF (border ownership) stimulation conditions. *Figure 3A* shows an example recording session (M1, *pen1*) in which 159 visually responsive neurons were recorded during CRF stimulation, and CCGs were computed for

7,956 pairwise combinations of neurons (Methods). CCGs were determined to be significant if the jitter-corrected CCG peak occurred within 10ms of zero-time lag and exceeded 7 standard deviations (SD) above the mean of the noise distribution (*Siegle et al., 2021*; *Trepka et al., 2022*; Methods). In this recording, 13.8% of the total CCGs computed were found to be significant. Eight significant pairs are highlighted in *Figure 3A*, with their corresponding CCGs shown in *Figure 3B*. The peak lag (PL), defined as the relative time delay in occurrence of peak correlation between two spike trains, was restricted to within 10ms. This metric approximates the synchrony and/or the direction of information flow between neuronal pairs (*Aertsen and Gerstein, 1985*; *Moore et al., 1970*; *Melssen and Epping, 1987*; *Ostojic et al., 2009*). These examples illustrate the tendency of CCGs to corroborate the circuitry within V1 columns under CRF conditions in that (1) PLs generally increase with larger distances between pairs, and (2) layer 4 neurons tend to lead superficial neurons (*Trepka et al., 2022*). We exploited this latter point to test the extent to which the sequence of information flow across layers differs between CRF and nCRF conditions.

## Comparison of cross-correlations during CRF and nCRF stimulation

From the 677 and 621 neurons recorded across the four recording sessions during CRF and nCRF stimulation, there were 41,759 and 28,430 pairwise combinations included, respectively (Methods). Of those, we found that 6937 (16.6%) and 4826 (17.0%) of the CCGs were significant (CRF: *pen1*: 1099/7956=13.8%, *pen2*: 2402/11431=21.0%, *pen3*: 2174/17780=12.2%, *pen4*: 1262/4592=27.5%; nCRF: *pen1*: 447/2473=18.1%, *pen2*: 2750/11823=23.3%, *pen3*: 839/6998=12.0%, *pen4*: 790/7136=11.1%). Notably, the proportion of significant connections was nearly identical under the two stimulus conditions and also was within the range observed in previous studies of macaque V1 (*Chu et al., 2014*; *Hembrook-Short et al., 2019*; *Smith and Kohn, 2008*; *Kohn and Smith, 2005*).

As noted above, modulation of visual responses during nCRF stimulation, such as border ownership stimuli, is thought to be generated principally by feedback/horizontal connections arriving within superficial and deep layers, which contrasts with feedforward input driving CRF stimulation (*Figure 3C*). Thus, we asked whether cross-correlations between pairs of neurons situated across laminar compartments depended on the type of visual stimulation. We focused our analysis on CCGs computed from pairs of neurons in which one neuron was located within laminar compartments receiving feedback/horizontal inputs ($FH_i$), namely layers 2/3 and 5/6, and the other was located within compartments relatively devoid of those inputs, namely 4C and 4A/B. This analysis compared the relative spike timing between neuronal pairs during CRF and nCRF stimulation. If indeed $FH_i$ contributes more to activity during the latter than the former, then we should expect this to be reflected in lead-lag relationships of the CCGs.

A comparison of V1 CCGs from key interlaminar neuronal pairs during CRF and nCRF stimulation is shown for an example recording session (M1, *pen1*) in *Figure 3D*. During CRF stimulation, the timing of spikes within compartments 4C and 4A/B generally preceded, or was simultaneous with, spikes within $FH_i$ compartments (CRF: 4C-2/3$_{PL}$ = 2ms, $p<10^{-7}$; 4A/B-2/3$_{PL}$ = 2ms, $p<10^{-28}$; 4C-5/6$_{PL}$ = 0ms, p=0.52; 4 A/B-5/6$_{PL}$ = 0ms, p=0.64; Sign test against zero-median). In contrast, during nCRF stimulation, relative spike timing was shifted in favor of neurons in $FH_i$ compartments. Specifically, during nCRF stimulation, the timing of spikes within compartments 4C and 4A/B lagged or was simultaneous with spikes within $FH_i$ compartments (nCRF: 4C-2/3$_{PL}$ = 0 ms, p=0.80; 4A/B-2/3$_{PL}$ = 0 ms, p=0.70; 4C-5/6$_{PL}$ = -2.5 ms, p=0.11; 4A/B-5/6$_{PL}$ = 0 ms, p=0.56; Sign test against zero median). To further quantify the apparent shift in lead-lag relationships, we calculated an asymmetry index for each CCG. The asymmetry index is computed as the difference in the CCG integral on either side of zero-time lag, and summarizes the lead-lag relationship independent of the CCG peak (*Jia et al., 2022*; Methods). Consistent with the analyses of PL, we found that CCG asymmetries differed significantly between CRF and nCRF stimulation for each of the interlaminar comparisons. In each case, the distribution of asymmetries during nCRF stimulation shifted to favor neurons in $FH_i$ laminar compartments, when compared to CRF stimulation (4C-2/3: CRF$_{Asym}$ = 0.036, nCRF$_{Asym}$ = –0.004, $\Delta$=–0.04, $p<10^{-6}$; 4A/B-2/3: CRF$_{Asym}$ = 0.046, nCRF$_{Asym}$ = –0.005, $\Delta$=–0.051, $p<10^{-11}$; 4C-5/6: CRF$_{Asym}$ = –0.010, nCRF$_{Asym}$ = –0.044, $\Delta$=–0.034, p=0.0015; 4A/B-5/6: CRF$_{Asym}$ = 0.012, nCRF$_{Asym}$ = –0.028, $\Delta$=–0.04, $p<10^{-5}$; Wilcoxon rank-sum test). Thus, CCG data from this single recording session indicated that relative spike timing was earlier within $FH_i$ laminar compartments during nCRF stimulation than during CRF stimulation.

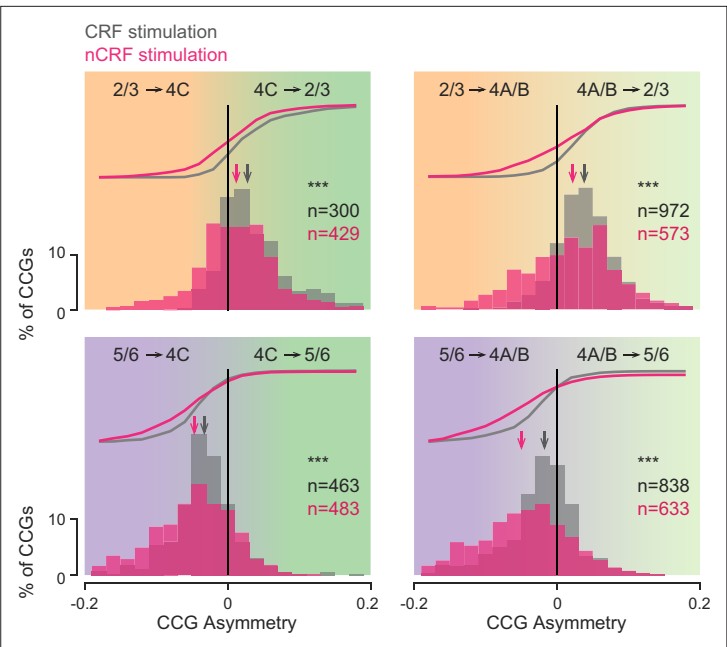

**Figure 4.** Comparison of interlaminar CCG asymmetries during CRF and nCRF stimulation. Combined dataset for each of the four key interlaminar combinations. Histograms and cumulative distributions of asymmetry for all significant CCGs during CRF (gray) and nCRF (magenta) stimulation. Arrows indicate the population median. ***p < 10^-5.

The online version of this article includes the following figure supplement(s) for figure 4:

**Figure supplement 1.** Comparison of interlaminar cross-correlations during CRF and nCRF stimulation under different CCG criteria.

---

Across all recordings, we identified a total of 2573 and 2118 significant neuronal pairs from the four interlaminar combinations during CRF (*pen1*=274; *pen2*=1187; *pen3*=560; *pen4*=302) or nCRF stimulation (*pen1*=210; *pen2*=1226; *pen3*=382; *pen4*=300), respectively, and compared their CCG asymmetries. Similar to the pattern observed in the example recording, we observed a consistent shift in lead-lag relationships for each of the interlaminar comparisons (*Figure 4*). Specifically, during nCRF stimulation, the distribution of asymmetries during nCRF stimulation shifted to favor neurons in FH$_i$ laminar compartments, when compared to CRF stimulation (4C-2/3: CRF$_{Asym}$ = 0.028, nCRF$_{Asym}$ = 0.012, $\Delta$=–0.016, p<10$^{-9}$; 4A/B-2/3: CRF$_{Asym}$ = 0.038, nCRF$_{Asym}$ = 0.022, $\Delta$=–0.016, p<10$^{-9}$; 4C-5/6: CRF$_{Asym}$ = –0.033, nCRF$_{Asym}$ = –0.047, $\Delta$=–0.014, p<10$^{-5}$; 4A/B-5/6: CRF$_{Asym}$ = –0.018, nCRF$_{Asym}$ = –0.05, $\Delta$=–0.032, p<10$^{-20}$; Wilcoxon rank-sum test). Thus, relative spike timing was earlier within FH$_i$ laminar compartments during nCRF stimulation than during CRF stimulation in the combined dataset, suggesting that the proportion of information flow from FH$_i$ layers to input layers increased during nCRF stimulation. This observation was consistent across individual recording sessions (*Supplementary file 2*). Notably, we observed a similar pattern when instead of asymmetry, we compared CCG peak lags (*Figure 4—figure supplement 1A*), and when we compared the total number of both significant and nonsignificant pairs (CRF = 15,920; nCRF = 11,263; *Figure 4—figure supplement 1B*), and when we compared CCGs with a more stringent significance criterion (10 SDs) in the combined dataset (*Figure 4—figure supplement 1C–D*).

Lastly, we asked whether interlaminar information flow was related specifically to border ownership modulation, independent of the comparisons between CRF and nCRF stimulation. Since CRF and nCRF stimuli differed in ways potentially unrelated to nonclassical modulation (e.g. variations within the CRF), we sought to assess the extent to which information flow from FH$_i$ to input layers predicted the magnitude of nCRF modulation. To this end, we computed two modulation indices from the same B$_{own}$ test stimuli (*Figure 2A*), specifically the B$_{own}$ modulation index and the local contrast (LC) index. In contrast to the B$_{own}$ modulation index, which reflects nCRF modulation, the LC index measures selectivity to luminance contrast polarity (dark-light vs. light-dark) within the CRF. We hypothesized that if

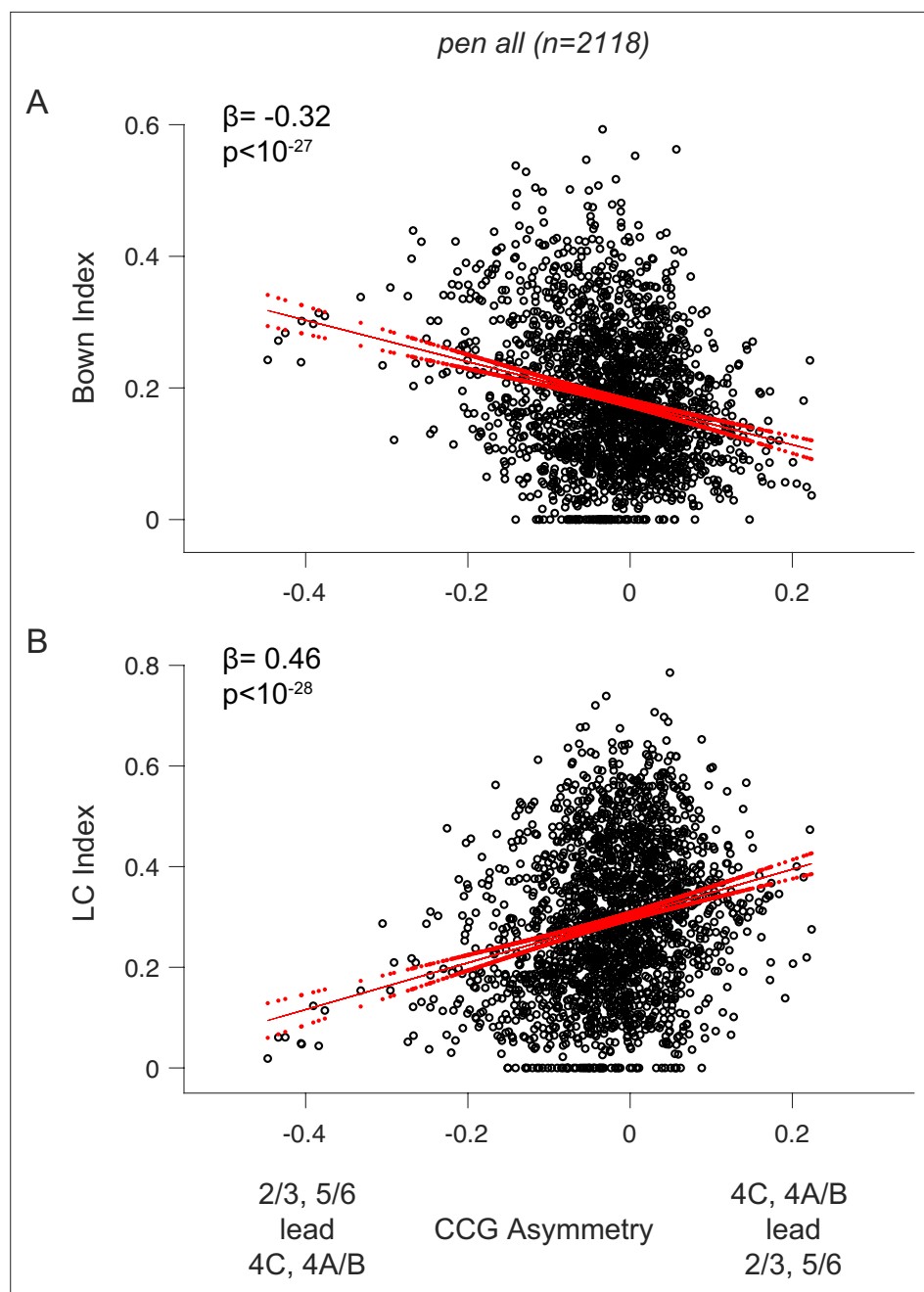

**Figure 5.** Dependence of nCRF modulation on interlaminar signal flow. (**A**) Scatter plot of the $B_{own}$ modulation index versus CCG asymmetry for all significant CCGs, aggregated across the four key interlaminar combinations and all recording sessions. CCG asymmetries were calculated during nCRF stimulation. The geometric mean of the $B_{own}$ index for each neuronal pair was used. Red lines denote the fitted regression line and the 95% confidence interval in linear regression. (**B**) Similar to (**A**), but using the local contrast (LC) index instead of the $B_{own}$ index. The LC index was calculated as the difference in neuronal responses to borders with opposite contrast polarities, normalized by the maximum response.

the relative timing of spiking activity within $FH_i$ laminar compartments is indicative of nCRF modulation, i.e., lower CCG asymmetries with respect to other compartments, to be associated with greater $B_{own}$. In contrast, we should not expect that relationship with the LC index.

Consistent with our hypothesis, we observed a negative relationship between the $B_{own}$ modulation index and CCG asymmetry across the population and within individual sessions (linear regression:

population slope (β)=–0.32, p<10$^{-27}$; individual sessions: *pen1*=–0.52, p<10$^{-5}$; *pen2*=–0.06, p=0.14; *pen3*=–0.25, p<10$^{-4}$; *pen4*=–0.20, p=0.008; *Figure 5A*). This result suggests that the magnitude of B$_{own}$ modulation for neuronal pairs depended on the proportion of information flow from FH$_i$ compartment to input layers. In contrast, the opposite was true for the LC index. That is, the LC index was positively correlated with CCG asymmetry across the population (linear regression: population slope (β)=0.46, p<10$^{-28}$; *Figure 5B*). However, this relationship was not consistent across sessions (individual sessions: *pen1*=–0.30, p=0.09; *pen2*=0.33, p<10$^{-5}$; *pen3*=0.23, p=0.006; *pen4*=0.14, p=0.04). To explore both results further, we built generalized linear models (GLMs) to assess the dependence of B$_{own}$ or LC modulation on CCG asymmetry to incorporate variations across interlaminar combinations and recording sessions. For LC modulation, this analysis indicated that the magnitude of LC modulation was not reliably predicted by interlaminar information flow (GLM: coefficient = –0.04, p=0.30; R$^2$=0.27; *Supplementary file 3*). In contrast, for B$_{own}$ modulation, this analysis confirmed that the magnitude of modulation was reliably predicted by interlaminar information flow, independent of variations across interlaminar combinations and recording sessions (GLM: coefficient = –0.16 p=1.1*10$^{-6}$; R$^2$=0.11). These results demonstrate that interlaminar information flow was related specifically to the magnitude of border ownership modulation.

## Discussion

We measured the sensitivity of neurons to nCRF stimulation, specifically border ownership, within large, local populations in macaque V1. Consistent with previous studies on awake animals, we observed V1 neurons with border ownership selectivity even under anesthesia. We observed that neurons within single columns tended to prefer the same side of objects located outside of the CRF. In addition, we found that cross-correlations among pairs of neurons situated across FH$_i$ and other laminar compartments differed between CRF and nCRF stimulation. Specifically, neurons within FH$_i$ laminar compartments were more likely to lead pairwise interactions with neurons in other compartments during nCRF stimulation than during CRF stimulation. Moreover, the magnitude of border ownership modulation was predicted by greater information flow from FH$_i$ laminar compartments, independent of the comparisons between CRF and nCRF stimulation. These results demonstrate that the flow of signals between neurons in FH$_i$ and other laminar compartments depends on the degree to which those neurons integrate visual information from beyond the CRF. Below we discuss some potential limitations of the results as well as their implications for the mechanisms underlying contextual modulation.

The observation that neurons within FH$_i$ laminar compartments can lead those in layer 4 during nCRF stimulation may appear surprising. However, several anatomical pathways could mediate the propagation of B$_{own}$ signals from FH$_i$ compartments to layer 4. In macaque V1, layers 5/6 send dense projections to 4A/B (*Lund, 1988*; *Yabuta and Callaway, 1998*). In particular, layer 6 pyramidal neurons, especially the subset classified as Type 1 cells, project substantially to layer 4C (*Wiser and Callaway, 1996*; *Fitzpatrick et al., 1985*). Projections from layers 2/3–4A/B have also been reported (*Blasdel et al., 1985*; *Callaway and Wiser, 1996*), and neurons in 4A/B often extend apical dendrites into layers 2/3 (*Lund, 1988*; *Yoshioka et al., 1994*). Although direct projections from layers 2/3–4 C are generally sparse (*Yabuta and Callaway, 1998*), a subset of neurons in the lower part of layer 3 can give off collateral axons to 4C (*Lund and Yoshioka, 1991*). Additionally, some 4C neurons extend dendrites into 4B, enabling potential dendritic integration of inputs from more superficial layers (*Somogyi and Cowey, 1981*; *Mates and Lund, 1983*; *Yabuta and Callaway, 1998*). Moreover, layers 2/3 may influence 4C neurons disynaptically, without requiring dense monosynaptic connections. Importantly, while CCGs can suggest possible circuit arrangements, functional connectivity may arise through mechanisms not fully captured by traditional anatomical tracing. Our observation of functional inputs from layers 2/3 to layer 4 aligns with prior findings in rodent V1, where CCG analysis (*Senzai et al., 2019*) or photostimulation (*Xu et al., 2016*) revealed similar pathways. Nonetheless, future studies will be needed to determine the extent of input to layer 4 neurons from more precise FHi laminae.

Each of the laminar compartments designated in this study comprised more than a single distinct anatomical layer. This was done to mitigate the uncertainty in determining exact laminar boundaries, to achieve comparable numbers of neurons across compartments, and to maximize statistical power. In addition, combining layers allowed us to combine feedback with horizontal input layers. It is known

that feedback and horizontal connections exhibit layer specificity within superficial and deep layers, for example horizontal connections are most prominent in layer 2/3 and 5 (*Rockland and Lund, 1983*; *Gilbert and Wiesel, 1983*), where feedback connections target primarily in layers 1, upper layer 2, and 5/6 (*Rockland and Pandya, 1979*; *Federer et al., 2021*; *Shmuel et al., 2005*). Yet, they both largely avoid layer 4 (*Angelucci et al., 2017*). Nonetheless, given that the number of anatomically distinct layers within macaque V1 well exceeds the four designated laminar compartments used here, it is important to consider the extent to which collapsing layers limits the conclusions one can draw from our observations. For example, although layer 6 receives feedback input, it also receives feedforward input directly from the LGN (*Hubel and Wiesel, 1972*; *Hendrickson et al., 1978*; *Blasdel and Lund, 1983*; *Yabuta and Callaway, 1998*). In addition, layer 4 A is generally considered an input layer whereas 4B is not (*Yabuta and Callaway, 1998*; *Hubel and Wiesel, 1972*; *Blasdel and Lund, 1983*; *Hendrickson et al., 1978*). Thus, the designation of any compartment as nominally input or feedback was relative and not absolute. That is, superficial (layers 2/3) and deep (layers 5/6) layers receive higher proportions of feedback/horizontal inputs than the 4A/B and 4C compartments. Indeed, previous studies of primate visual cortex have used a similar approach in comparing input and feedback/horizontal laminar compartments (e.g. *Hansen et al., 2012*; *Nandy et al., 2017*; *van Kerkoerle et al., 2017*; *Pettine et al., 2019*). Thus, the observed changes in pattern of cross-correlations between neurons situated across compartments with different relative feedforward and feedback/horizontal input can be interpreted in light of those input differences.

Another important consideration is that a key challenge of studying classical and nonclassical RF effects simultaneously in large neuronal populations is the difficulty in achieving alignment of all CRFs with the test stimuli. Although our probe penetrations were largely normal to the surface and neurons showed minimal deviations in orientation selectivity across the cortical depth, V1 neurons nonetheless exhibit noteworthy RF scatter (location/size) even within single cortical columns (*Tootell et al., 1988*; *Gur et al., 2005*; *Li et al., 2022*). This means that the positioning of stimuli relative to the CRF inevitably varied across simultaneously recorded neurons (e.g. *Figure 1C*). Moreover, this means that although neurons were generally well driven by CRF grating stimuli and exhibited clear orientation tuning, CRF stimulation was not guaranteed to fall solely within the CRF of all neurons. Nonetheless, involvement of the surround, especially far surround, was considerably less extensive for CRF stimulation than for nCRF stimulation. Furthermore, in the latter case, regardless of variability in the CRF location, stimuli falling within the CRF were always matched across nCRF stimulus conditions, and the surrounds were also identical within 8° of the CRF (*Figure 2A and B*).

The experiments described here were performed in anesthetized animals. Several other types of contextual modulation have been demonstrated in anesthetized animals (*Ramsden et al., 2001*; *Hupé et al., 1998*; *Bair et al., 2003*; *Allman et al., 1985*; *Gilbert and Wiesel, 1990*; *Nothdurft et al., 1999*; *Webb et al., 2003*; *Zarella and Ts'o, 2016*; *Müller et al., 2003*; *Henry et al., 2013*; *Bijanzadeh et al., 2018*), and can involve inter-cortical feedback (*Nurminen et al., 2018*). Nonetheless, anesthesia is known to affect certain types of contextual responses in area V1. For example, the 'figure-ground segregation' effect, where neurons respond more strongly to a texture in a 'figure' region than to the same texture in a 'ground' region, has been shown to be suppressed under anesthesia (*Lamme et al., 1998*) and when figure stimuli are not perceived (*Supèr et al., 2001*). However, there has been debate over whether this effect depends on attention and awareness (*Marcus and Van Essen, 2002*; *Jones et al., 2015*; *Poltoratski and Tong, 2020*; *Poort et al., 2012*; *Huang et al., 2020*). It is worth noting that in that case, RFs are centered on the figure regions, and the modulation likely involves detecting feature discontinuities around boundaries, followed by region-filling and background suppression. Each step could be influenced differently by behavioral variables. Nevertheless, the center figure enhancement emerges late in the V1 response, ~55 ms after response onset (*Poort et al., 2012*). In contrast, border ownership involves RFs centered on the boundaries, with modulation emerging much earlier at ~10–35 ms after response onset (*Sugihara et al., 2011*). Therefore, although both forms of modulation may involve feedback/horizontal input, the underlying circuitry may differ. Our finding of border ownership modulation in anesthetized animals aligns with previous studies showing that border ownership modulation can act separately even without attention (*Qiu et al., 2007*; O'Herron and *O'Herron and von der Heydt, 2009*) or before object shape recognition (Williford and *Williford and von der Heydt, 2016*; Ko and *Ko and von der Heydt, 2018*), perhaps signifying a representation of 'proto-objects' in the early visual cortex through

automatic, pre-attentive grouping mechanism (*Martin and von der Heydt, 2015*; *von der Heydt, 2023*; *Self et al., 2019*).

Border ownership enables neurons with small CRFs in the early visual cortex to assign the occluding border between image regions to a foreground object, which is crucial for natural scene segmentation and object recognition (*Nakayama et al., 1995*). Human imaging studies have demonstrated the existence of $B_{own}$ in both lower (*Fang et al., 2009*) and higher visual areas (*Kourtzi and Kanwisher, 2001*; *Andrews et al., 2002*), yet the underlying circuitry is still poorly understood. Several computational models have been proposed to explain $B_{own}$, including feedforward models (*Walker et al., 1999*; *Sakai and Nishimura, 2006*; *Supèr et al., 2010*), horizontal models (*Zhaoping, 2005*; *Kogo et al., 2010*), and feedback models (*Craft et al., 2007*; *Jehee et al., 2007*; *Jeurissen et al., 2016*). Yet cues for determining $B_{own}$ often lie far from the CRF and beyond the extents of geniculocortical and horizontal V1 connections (*Angelucci et al., 2017*). Furthermore, conduction along horizontal fibers appears too slow (0.1–0.4 mm/ms; *Grinvald et al., 1994*; *Bringuier et al., 1999*; *Girard et al., 2001*) to account for the rapid (~10–35 ms) emergence of $B_{own}$ and its independence from object size (Zhang and *Zhang and von der Heydt, 2010*; *Sugihara et al., 2011*). In contrast, feedback inputs can be conducted through fibers 10 times faster (20–60 mm/ms; *Girard et al., 2001*), even via monosynaptic connections (*Siu et al., 2021*), and contribute to modulating the early responses of V1 neurons (*Hupé et al., 2001*; *Bair et al., 2003*). Thus, although we combined feedback and horizontal recipient layers, the differences in interlaminar signal flow we observed were likely driven predominantly by feedback inputs.

In primates, V1 receives inter-cortical feedback connections primarily from areas V2, V3, V4, and MT (*Rockland et al., 1994*; *Markov et al., 2014*). Thus, V1 $B_{own}$ signals could be generated via inputs from these areas (*Zhou et al., 2000*; *Hesse and Tsao, 2016*; *Hesse and Tsao, 2023*; *Franken and Reynolds, 2021*; *Zhu et al., 2020*). Our results resonate with a recent study reporting that in area V4, $B_{own}$ emerges earliest in deep layers compared to input layers (*Franken and Reynolds, 2021*). This suggests that the earliest component of $B_{own}$ in V4 is not inherited from feedforward inputs, that is from V1/V2, but is generated within deep layers de nova, or through feedback connections that arrive at deep layers (*Franken and Reynolds, 2021*). This further supports the view that $B_{own}$ in V1 emerges via feedback from higher visual areas. In addition, similar to what we observed in V1, columnar organization of $B_{own}$ was observed in V4 (*Franken and Reynolds, 2021*). Indeed, neurons with $B_{own}$ are clustered into patches in a wide range of primate visual areas, including areas V2, V3, V3A, V4, and V4A (*Hesse and Tsao, 2023*). Thus, the evidence of $B_{own}$ modularity we observed in V1 could arise from organized feedback from the $B_{own}$ modules in those areas. This is supported by recent anatomical evidence suggesting that feedback terminals in V1 are clustered and functionally specific (*Angelucci et al., 2002*; *Federer et al., 2021*; *Shmuel et al., 2005*; *Siu et al., 2021*).

## Methods

### Experimental model and subject details

Anesthetized recordings were conducted in two adult male rhesus macaques (*Macaca mulatta*, M1, 13 kg; M2, 8 kg). All experimental procedures were in accordance with the National Institutes of Health Guide for the Care and Use of Laboratory Animals, the Society for Neuroscience Guidelines and Policies, and with approved Institutional Animal Care and Use Committee (IACUC) protocol (#APLAC-9900) of Stanford University.

### Electrophysiological recordings

Prior to each recording session, treatment with dexamethasone phosphate (2 mg per 24 hr) was instituted 24 hr to reduce cerebral edema. After administration of ketamine HCl (10 mg per kilogram body weight, intramuscularly), monkeys were ventilated with 1–2% isoflurane in a 1:1 mixture of $N_2O$ and $O_2$ to maintain general anesthesia. Electrocardiogram, respiratory rate, body temperature, blood oxygenation, end-tidal $CO_2$, urine output, and inspired/expired concentrations of anesthetic gases were monitored continuously. Normal saline was given intravenously at a variable rate to maintain adequate urine output. After a cycloplegic agent (atropine sulfate, 1%) was administered, the eyes were focused with contact lenses on an LCD monitor. Vecuronium bromide (60 µg/kg/hr) was infused to prevent eye movements.

With the anesthetized monkey in the stereotaxic frame, an occipital craniotomy was performed over the opercular surface of V1. The dura was reflected to expose a small (~3 mm$^2$) patch of cortex. Next, a region relatively devoid of large surface vessels was selected for implantation, and the Neuropixels probe was inserted with the aid of a surgical microscope. Given the width of the probe (70 μm x 20 μm), insertion of it into the cortex sometimes required multiple attempts if it flexed upon contacting the pia. The junction of the probe tip and the pia could be visualized via the (Zeiss) surgical scope, and the relaxation of pia dimpling was used to indicate penetration, after which the probe was lowered at least 3–4 mm. Prior to probe insertion, it was dipped in a solution of the DiI derivative FM1-43FX (Molecular Probes, Inc) for subsequent histological visualization of the electrode track.

Given the length of the probe (1 cm), and the complete distribution of electrode contacts throughout its length, recordings could be made either in the opercular surface cortex (M1, *pen* 1–2) or within the underlying calcarine sulcus (M2, *pen* 3–4), by selecting a subset of contiguous active contacts (n=384) from the total number (n=986). Recordings were made at 1–2 sites in one hemisphere of each monkey. At the end of the experiment, monkeys were euthanized with pentobarbital (150 mg/kg) and perfused with normal saline followed by 1 liter of 1% (wt/vol) paraformaldehyde in 0.1 M phosphate buffer, pH 7.4.

## Visual stimulation

Visual stimuli were presented at manually mapped receptive field (RF) locations for each recording session on an LCD monitor (Model NEC-4010, dimensions: 88.5 cm (H) x 49.7 cm (V), resolution: 1360x768 pixels, frame rate: 60 Hz) positioned 114 cm from the monkeys. RF eccentricities were ~4–6° (M1) and ~6–10° (M2). Visual stimuli were generated using customized MATLAB scripts with the Psychophysics Toolbox extensions (version PTB-3; *Kleiner et al., 2007*). A photodiode was used to measure stimulus timing.

CRF stimuli consisted of drifting Gabor gratings (2°/s, 100% Michelson contrast) with a diameter of 1.5 dva, positioned within the joint receptive fields (RFs) of recorded neurons. This size was selected to largely constrain the stimuli within the CRFs of recorded neurons, typically ~0.5–1 dva, as determined manually online. Gratings drifted in 36 different directions, from 0 to 360° in 10° steps, in a pseudo-random order. Each stimulus condition was presented for 1 s and repeated 5 or 10 times. A blank screen with equal luminance to the Gabor patch was presented for 0.25 s during the stimulus interval. Stimuli were presented either monocularly (*pen* 1, 3) or both monocularly and binocularly (*pen* 2, 4). Four spatial frequencies (0.5, 1, 2, 4 cycles/°) were tested. The optimal eye and spatial frequency conditions were determined offline for further analysis.

nCRF stimuli used in this study, that is Border ownership test stimuli consisted of uniformly white or black objects (square, 8x8 dva) presented on a black or white background, respectively, like previous studies (*Zhou et al., 2000*). One border of the object fell within the CRFs of the recorded neurons at an orientation that varied in 45° steps. Objects were either 4x4 dva or 8x8 dva in size, and only conditions with 8x8 dva were selected for quantifying border ownership selectivity to tolerate the variation in the CRF locations. Stimuli were presented monocularly to the preferred eye, to avoid the surround stimulating the CRF of the non-preferred eye. For each of the four orientation conditions, there were four basic conditions (*Figure 2A*). Within the CRF, the oriented border could either be a dark-light edge or light-dark edge, resulting in two different local contrast (LC) polarity conditions, namely, LC1 and LC2. For the same LC condition, the stimulus configuration outside of the CRF differed such that the oriented border could belong to opposite sides of objects, resulting in two different sides of object conditions, namely, side 1 and side 2. Each stimulus condition was presented for 1 s and repeated 10 or 20 times. A blank screen with equal luminance to the mean of object and background luminance was presented for 0.5 s during the stimulus interval.

## Receptive field mapping

In some sessions, the receptive fields of V1 neurons were tested systematically for offline use. CRFs were mapped from a 10x10 array (4x4 dva) using probe stimuli. The array was centered at the manually determined RF location. The stimuli consisted of circular sine wave gratings of 0.4 dva in diameter and 100% Michelson contrast. Gratings were presented at the optimal orientation determined manually online for each recording session and drifted at a speed of 2°/s. Each stimulus condition

was presented for 0.2 s, interleaved with 0.3 s of blank screen with equal luminance, and repeated 15 times.

Mean spike counts from 0.05 to 0.25 s after each stimulus onset were calculated, resulting in a 10x10 matrix containing mean spike rates to each grid location. This 2D matrix was then interpolated to a 40x40 matrix (MATLAB function 'interp2') and then smoothed with a 2D Gaussian filter (MATLAB function 'imgaussfilt', σ=2). The outline of the CRF was defined as the contour isoline at 80% of the maximum response using MATLAB function 'contour3'. Neurons with only 1 contour region at the threshold were selected.

### Data acquisition and spike sorting

Raw spike-band data were sampled and recorded at 30 kHz. They were then median-subtracted and high-pass filtered at 300 Hz during the pre-processing stage. Spike sorting was carried out with Kilosort 3 (*Pachitariu et al., 2024*) to find spike times and assign each spike to different templates (neurons). Default parameters in Kilosort 3 were used for spike sorting, specifically, Ops.th = [9,9]; Ops.lam=20; Ops.AUCsplit=0.8; Ops.ThPre=8; Ops.sig=20; Ops.nblocks=5; Ops.spkTh = –6. The raw sorted data were then manually curated in Phy (https://github.com/cortex-lab/phy; *Rossant et al., 2025*) to remove templates with very few spikes or with atypical waveforms and to perform minimal templates merging and splitting. Double-counted spikes from the algorithm fitting the residuals to a new template were identified using established criteria (*Siegle et al., 2021*), by counting spikes from two templates separated by less than 50 μm (~5 channels) and with spike activity occurring within five samples (0.167ms). Double-counted spikes from the template smaller in amplitude were removed from further analysis. Visual responsiveness of each unit was assessed under CRF, nCRF, or RF mapping stimulation. For each type of stimulation, a paired-sample t test was performed to determine whether the mean spike counts after each stimulus presentation exceeded the preceding blank period at a significance level of 0.01. Only neurons showing significant visual responsiveness were included for further analysis.

### Layer assignment

The laminar locations of our recorded neurons were estimated based on a combination of functional analysis and histology results (*Zhu et al., 2024*; *Carr et al., 2025*). For each recording, we first performed the current source density (CSD) analysis on the stimulus-triggered average of local field potentials (LFPs). LFPs were low-pass filtered at 200 Hz and recorded at 2500 Hz. LFP signals recorded from every four nearby channels were averaged and realigned to the onset of visual stimulus. CSD was estimated as the second-order derivatives of signals along the probe axis using the common five-point formula (*Nicholson and Freeman, 1975*). The result was then smoothed across space (σ=120 μm) to reduce the artifacts caused by varied electrode impedance. We located the lower boundary of the major sink (the reversal point of sink and source) as the border between layer 4C and layer 5/6. We also considered anatomical data in order to localize recorded neurons within gray matter, allowing for minor adjustments (±1 group channel) in layer boundary placement. Subsequent layer boundaries were determined by offsetting the cortical thickness derived from histological slices (*Figure 1—figure supplement 1*).

### Single neuron properties during CRF stimulation

To assess the orientation tuning for each neuron, the spike rates during 0.1–1 s after each stimulus onset were calculated and averaged for each orientation condition (N=18). The orientation tuning responses were first smoothed with a Hanning window (half-width at half-height of 20°), and then fitted with a von Mises function (*Swindale, 1998*)

$$y = a_0 + a_1 * e^{a_2 * (\cos(2*x - 2*a_3) - 1)}$$

The location of the peak of the fitted curve was determined as the preferred orientation. Only neurons well fit by the function ($R^2 > 0.7$) were included for determining pairwise differences in preferences for all combinations in each recording session.

### Modulation index for border ownership

As in previous studies (*Zhou et al., 2000*), we calculated a $B_{own}$ modulation index to quantify the selectivity of each neuron for $B_{own}$. The $B_{own}$ modulation index is defined as

$$B_{own}index = \frac{0.5 * (R_{LC1} + R_{LC2})_{Side1} - 0.5 * (R_{LC1} + R_{LC2})_{Side2}}{R_{max}}$$

Here, $R_{LC, Side}$ represents the average spike rate from 0.05 to 0.5 s after stimulus onset for one of four basic conditions, which are derived from the combination of two different *Local Contrast* conditions (LC1 and LC2) and two *Side* conditions (Side 1 and Side 2). Baseline activity was not subtracted. Since the border within the CRF can vary in 45° steps from 0° to 180°, we only selected the optimal orientation for each recording population to calculate the $B_{own}$ index. $R_{max}$ was defined as the maximum response for each neuron across the four basic conditions and four orientations. We used $R_{max}$ instead of summing responses across the four basic conditions to account for the slight variations in orientation preferences among neurons within each recording session. The statistical significance of $B_{own}$ was evaluated using a two-factor ANOVA (MATLAB function 'anovan') on the average spike rates for each condition. The two factors were object *Side* (with two levels: Side 1 and Side 2) and *Local Contrast* (with two levels: LC 1 and LC 2), including their interactions. A significance level of 0.05 was used. Neurons were considered to exhibit significant $B_{own}$ modulation if they showed a significant main effect for the object *Side* factor. Whether they were invariant to local contrast was determined by the main effect of the *Local Contrast* factor.

## Cross-correlograms (CCGs)

To measure correlated firing, we computed the cross-correlations between spike trains of all pairs of simultaneously recorded neurons. For each recording session, we calculated cross-correlograms (CCGs) from pairwise combinations of neuronal spike trains during either CRF (Gabor gratings) or nCRF (border ownership) stimulation conditions, respectively. We only included neuronal pairs with geometric mean firing rates more than 0.5 spikes/sc for further analysis. For both conditions, we focused on the spiking activity within the 0.25–1 s window during each stimulus presentation to mitigate the influence of the transient visual response after stimulus onset. The CCG function for a given pair of neurons was defined as follows:

$$CCG(\tau)_{j-k} = \frac{\frac{1}{M} \sum_{i=1}^{M} \sum_{t=1}^{N-\tau} x_j^i(t) \times x_k^i(t+\tau)}{\theta(\tau) \sqrt{\lambda_j \lambda_k}}$$

where $j$ denotes the first/reference neuron and $k$ denotes the second/target neuron in the CCG function. $\tau$ is the time lag between the two spike trains $x_j^i$ and $x_k^i$ from neuron $j$ and $k$, during trial $i$. The value of $x_j^i(t)$ and $x_k^i(t)$ is 1 if there is a spike at time bin $t$ and is zero otherwise. $M$ is the total number of trials; $N$ is the number of time bins within a trial. $\theta(\tau) = N - |\tau|$, is a triangular function that corrects for the degree of overlap in the two spike trains at each time lag. $\lambda_j$ and $\lambda_k$ are the mean firing rates of neuron $j$ and $k$ computed over the same bins used to compute the CCG at each time lag. The normalization by the geometric mean of spike rates is commonly used by previous studies to account for the influence of firing rates on the CCG peaks (**Bair et al., 2001**; **Kohn and Smith, 2005**).

To correct for correlations due to stimulus-locking or slow fluctuations in the population responses (e.g. gamma-band activity), we computed a jitter-corrected CCG by subtracting a jittered CCG from the original CCG:

$$CCG_{corrected} = CCG_{original} - CCG_{jittered}$$

The jittered CCG ($CCG_{jittered}$) reflects the expected value of CCG computed from all possible jitters of each spike train within a given jitter window (**Harrison and Geman, 2009**; **Smith and Kohn, 2008**). The jittered spike train preserves both the PSTH of the original spike train across trials and the spike counts in the jitter window within each trial. As a result, jitter correction removes the correlation between PSTHs (stimulus-locking) and correlation on timescales longer than the jitter window (slow population correlations). Here, a 25 ms jitter window was chosen based on previous studies (**Jia et al., 2013**; **Siegle et al., 2021**). Jitter-corrected CCG ($CCG_{corrected}$) was then smoothed with a 5ms kernel (0.05, 0.25, 0.4, 0.25, 0.05; **Kohn and Smith, 2005**; **Smith and Kohn, 2008**) before further analysis.

We classified a CCG as significant if the peak of the jitter-corrected CCG occurred within 10ms of zero and exceeded 7 standard deviations (SD) above the mean of the noise distribution, similar to previous studies (**Siegle et al., 2021**). The noise distribution for a CCG was defined as the flanks of

the jitter-corrected CCG ($\{CCG\left(\tau\right)|100 \geq |\tau| \geq 50ms\}$). All analyses presented in the main text involve only significant, jitter-corrected CCGs. In the supplementary material, we also applied a more stringent criterion, requiring that significant CCG peaks exceed 10 SD above the noise, and a less stringent criterion that includes both significant and non-significant CCGs. These ensure that our results are robust across the choices of threshold (*Figure 4—figure supplement 1*).

To assess the directionality of functional interactions or signal flow between pairs of neurons, we focused on two metrics derived from the CCG functions. One is the peak lag (PL), which describes the time lag of peak correlation between the spike trains of neuron $k$ (target) relative to neuron $j$ (reference). PL is positive if the reference neuron $j$ leads target neuron $k$ in spike timing, negative if $j$ lags $k$, and zero if j and k fire synchronously. The second metric we used to assess the signal flow is CCG asymmetry, computed by subtracting the sum of CCG values during the [–13, 0] ms time window from the sum of CCG values during the [0, 13] ms. Similar to PL, a positive CCG asymmetry indicates that reference neuron $j$ leads $k$, while a negative asymmetry indicates that $j$ lags $k$. The 13ms window was chosen as half of the 25ms jitter window, consistent with previous studies (*Jia et al., 2022*). Compared to peak lag, CCG asymmetry additionally captures the strength of the functional interactions between pairs of neurons and is less dependent on the exact shape of the CCG peak, which is often more complex in corticocortical functional interactions (*Alonso and Martinez, 1998*).

### Relationship between $B_{own}$ modulation and interlaminar signal flow

To determine whether the magnitude of border ownership or local contrast modulation is predicted by interlaminar signal flow during nCRF stimulation, we first performed linear regression (MATLAB, 'fitlm', ordinary least squares) and then built generalized linear models (GLMs) using predictors including CCG asymmetry, interlaminar combinations, and recording sessions. We focused on the four key interlaminar combinations (4C-2/3, 4A/B-2/3, 4C-5/6, 4A/B-5/6), where the latter laminar compartments receive feedback/horizontal inputs (FH$_i$). Only neuronal pairs with significant CCGs were included as individual samples in the GLM. The resulting GLM equations were:

$$B_{own} \sim \left(pair\right)\ 1 + asymmetry + laminar\ combination + pen$$
$$LC \sim \left(pair\right)\ 1 + asymmetry + laminar\ combination + pen$$

$B_{own}$ (pair) and LC (pair) were calculated as the geometric mean of the $B_{own}$ and LC indices for the constituent neurons in each pair. Specifically, $B_{own}\left(pair\right) = \sqrt{abs(B_{own}(j)) * abs(B_{own}(k))}$, where abs($B_{own}(j)$) and abs($B_{own}(k)$) represent the absolute $B_{own}$ modulation indices for constituent neurons $j$ and $k$. A similar calculation was performed for LC (pair). CCG asymmetries were calculated from the nCRF stimulation condition where neurons in the input layers were selected as reference neurons and the ones in FH$_i$ compartments were target neurons in the CCG. Additional predictors, including interlaminar combinations and recording sessions, were included in the models to incorporate variations when pooling the dataset.

### Statistical tests

$B_{own}$ for each neuron was assessed with ANOVA. The proportion of $B_{own}$ neurons within different layers was assessed using chi-squared test of homogeneity. The sign of populational border ownership preferences from individual recording sessions was tested with a sign test. The sign of populational CCG peak lag for a given laminar compartment combination was assessed with a sign test. The comparisons of peak lag and CCG asymmetry between CRF and nCRF stimulation conditions for each laminar compartment combination were evaluated using Wilcoxon rank-sum tests. The dependence between CCG asymmetry and $B_{own}$ for a pair of neurons was assessed using linear regression and GLM.

Combined dataset for each of the four key interlaminar combinations. Histograms and cumulative distributions of asymmetry for all significant CCGs during CRF (gray) and nCRF (magenta) stimulation. Arrows indicate the population median. ***p<10$^{-5}$.

### Acknowledgements

We thank Jonathan C Horton for extensive help with the recordings and histology, Danielle A Lopes, Stephen Cital, Shellie Hyde and Sam Baker for technical assistance, Tim Harris and Karel Svoboda for providing the Neuropixels probes. This work was supported by NIH Grant EY029759, Brain and

Behavior Research Foundation Grant to XM; NIH Grants EY014924, NS116623, and a Ben Barres Professorship to TM.

# Additional information

## Competing interests

Tirin Moore: Senior editor, eLife. The other authors declare that no competing interests exist.

## Funding

| Funder | Grant reference number | Author |
|---|---|---|
| National Eye Institute | EY014924 | Tirin Moore |
| National Eye Institute | EY026877 | Tirin Moore |
| National Institute of Neurological Disorders and Stroke | NS116623 | Tirin Moore |
| National Eye Institute | EY029759 | Xiaomo Chen |

The funders had no role in study design, data collection and interpretation, or the decision to submit the work for publication.

## Author contributions

Shude Zhu, Conceptualization, Data curation, Software, Formal analysis, Validation, Investigation, Methodology, Writing – original draft, Writing – review and editing, Visualization; Yu Jin Oh, Formal analysis; Ethan B Trepka, Software, Formal analysis; Xiaomo Chen, Investigation, Methodology; Tirin Moore, Conceptualization, Validation, Investigation, Methodology, Writing – original draft, Writing – review and editing

## Author ORCIDs

Shude Zhu ⬚ https://orcid.org/0000-0002-8674-9607
Xiaomo Chen ⬚ https://orcid.org/0000-0002-6142-9981
Tirin Moore ⬚ https://orcid.org/0000-0002-3345-2930

## Ethics

All experimental procedures were in accordance with the National Institutes of Health Guide for the Care and Use of Laboratory Animals, the Society for Neuroscience Guidelines and Policies, and with approved Institutional Animal Care and Use Committee (IACUC) of Stanford University (IACUC) protocol (#APLAC-9900).

Reviewer #1 (Public review): https://doi.org/10.7554/eLife.103255.3.sa1
Reviewer #2 (Public review): https://doi.org/10.7554/eLife.103255.3.sa2
Reviewer #3 (Public review): https://doi.org/10.7554/eLife.103255.3.sa3
Author response https://doi.org/10.7554/eLife.103255.3.sa4

# Additional files

## Supplementary files

Supplementary file 1. Number of neurons selective to Bown (LC) across recordings.

Supplementary file 2. Comparison of CCG asymmetries during CRF and nCRF for each recording.

Supplementary file 3. GLMs for dependence of border ownership and local contrast on CCG asymmetry.

MDAR checklist

## Data availability

All raw data generated as part of this study are deposited at https://doi.org/10.5061/dryad.7m0cfxq94 and are publicly available. All the raw code developed for data analysis has been deposited to GitHub (https://github.com/szhu-007/zhu_v1_interlaminar-signal-flow copy archived at *Zhu et al., 2025*) and is freely available for access.

The following dataset was generated:

| Author(s) | Year | Dataset title | Dataset URL | Database and Identifier |
|---|---|---|---|---|
| Zhu S, Oh Y, Trepka E, Chen X, Moore T | 2025 | Dependence of contextual modulation in macaque V1 on interlaminar signal flow | https://doi.org/10.5061/dryad.7m0cfxq94 | Dryad Digital Repository, 10.5061/dryad.7m0cfxq94 |

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
