## [Editor Report · eLife Assessment]

The results by Zhu et al provide **valuable** insights into the representation of border ownership in area V1. They used neuropixel recording to demonstrate the clustering of border ownership, and compared cross-correlation functions between neurons in different layers to demonstrate that they depend on the type of stimulus. The strength of the evidence is **solid** but can be improved by performing additional analyses and addressing some concerns (as raised in the previous and current review), and accounting for the differences in classical and non-classical receptive field stimulation conditions.

---

## [Referee Report · Reviewer #1 (Public review)]

Zhu and colleagues used high-density Neuropixel probes to perform laminar recordings in V1 while presenting either small stimuli that stimulated the classical receptive field (CRF) or large stimuli whose border straddled the RF to provide nonclassical RF (nCRF) stimulation. Their main question was to understand the relative contribution of feedforward (FF), feedback (FB), and horizontal circuits to border ownership (*B_own_*), which they addressed by measuring cross-correlation across layers. They found differences in cross-correlation between feedback/horizontal (FH) and input layers during CRF and nCRF stimulation.

Comments on revisions:

In the revision, the authors have added a paragraph in the Discussion to address the question of layers 2/3 neurons leading layer 4 neurons, and have provided answers to the questions in the public review without making substantial changes in the paper. However, there were several other recommendations, which I am not sure why were not considered. I am adding those again below.

* For CRF stimulation, the zero lag between 4C and 4A/B with layer 5/6 (Figure 3D last two columns on the right) was surprising to me. I just felt that this could be because layer 6 may also be getting FF inputs. Perhaps better not to club layer 5 with 6, as mentioned earlier also.

* Interpreting the nCRF delays, with often negative delays, was very challenging for me. For example, 4C -> 5/6 (third column in Figure 3) has a significantly negative peak (although that does not show up in statistical analysis because it seems to be a signed test to just test if the median was greater than zero, not if the median was different from zero; line 285). What is the interpretation here? Are spikes in 5/6 causing spikes in 4C (which, as mentioned earlier, would require anatomical projections from 5/6 to 4C)? On the other hand, if FB inputs arrive in 5/6 but there are no inputs going to 4C, then why should there even be a significant cross-correlation?

The only explanation I could think of is somehow an alignment of inputs in these two layers such that FH inputs come in Layer 5/6 just before FF inputs arrive in 4C, each causing a spike in a neuron in each layer which are otherwise not anatomically interconnected. But this would require both a very precise temporal coupling between FF and FH inputs arriving in these areas AND neurons in layer 5/6 which very strongly respond to FH stimulation (I thought that FH inputs are mainly modulatory and not as strong). Anyway, it would be good to see some cross correlation functions which have a negative lag (all examples in Fig 3B has positive or zero lag).

* I think cross-correlation analysis would have been useful if there was data from a feedback area (say V2). In its absence, perhaps latency analysis (by just comparing the PSTH) could have revealed something interesting, given that the hypothesis is about differences in the timings in FH versus FF inputs. Do PSTHs across layers show the type of differences that are being claimed (e.g. in line 295-297)?

* Line 262-63: "Notably, the rates were nearly identical under the two stimulus conditions" - I would have thought CRF stimulation would produce higher rates. Can the authors explain this?

* Line 174-175: Isn't the proportion of border ownership cells in layer 4C higher than one would expect under the assumption that nCRF effects are mediated by horizontal and feedback connections which layer 4C does not receive? Can authors explain?

* Figure 3D: it would also be good to show the heatmaps stacked up in the increasing order of the interelectrode distance of the pairs so that it will be easy to see how the peak lag changes with distance as well.

* It will be good to show the shift in peak lag and CCG asymmetry between CRF and nCRF conditions for the same pairs, using a violin or bar plot with lines connecting each pair in Figure 3.

* Line 594, 603, 628 and 630: What procedure was used to determine the size, location of the CRF, and optimal orientation manually online?

* Line 733-734: Although a reference is cited, please explicitly mention the rationale for keeping the peak lag cutoff at 10 ms.

* It is unclear why a grating was used for the CRF condition, instead of just having the portion of the stimulus within the RF for the nCRF condition, as the comparisons for FHi with FF are with different FF drives in each case.

* Figure 5 - the scatter is enormous, can you please provide the R2 values?

---

## [Referee Report · Reviewer #2 (Public review)]

Summary:

The authors present a study of how modulatory activity from outside the classical receptive field (cRF) differs from cRF stimulation. They study neural activity across the different layers of V1 in two anesthetized monkeys using Neuropixels probes. The monkeys are presented with drifting gratings and border-ownership tuning stimuli. They find that border-ownership tuning is organized into columns within V1, which is unexpected and exciting, and that the flow of activity from cell-to-cell (as judged by cross-correlograms between single units) is influenced by the type of visual stimulus: border-ownership tuning stimuli vs. drifting-grating stimuli.

Strengths:

The questions addressed by the study are of high interest, and the use of Neuropixels probes yields extremely high numbers of single-units and cross-correlation histograms (CCHs) which makes the results robust. The study is well-described.

Comments on revisions:

The results are interesting and seem robust. However, several of my main points were not addressed. The authors do not analyze or discuss the problem the border ownership stimuli do uniquely isolate feedback from feedforward influences. Here are my remaining points/recommendations:

(1) In my previous review I indicated that the border-ownership signal also provides a strong feedforward drive, a black-white edge, in addition to the border ownership signal. Calling this a "nCRF stimulus" is a misnomer. Please correct this terminology and replace it by something that is appropriate, e.g. changing it into "grating stimulation" (instead of CRF stimulation) and BO-stimulation (instead of nCRF stimulation).

(2) In my previous review I asked if the initial response for the border ownership stimulus show the feedforward signature. It is unclear to me why this suggestions did not lead to an analysis of the feedforward response. I repeat the text from my previous review: "The authors state that they did not look at cross-correlations during the initial response, but if they do, do they see the feedforward-dominated pattern? The jitter CCH analysis might suffice in correcting for the response transient." Can the authors address this point?

(3) In my previous review I asked the authors show the average time course of the response elicited by preferred and nonpreferred border ownership stimuli across all significant neurons. It remains unclear why this plot was not provided.

---

## [Referee Report · Reviewer #3 (Public review)]

Summary:

The paper by Zhu et al is on an important topic in visual neuroscience, the emergence in the visual cortex of signals about figure and ground. This topic also goes by the name border ownership. The paper utilizes modern recording techniques very skillfully to extend what is known about border ownership. It offers new evidence about the prevalence of border ownership signals across different cortical layers in V1 cortex. Also, it uses pairwise cross correlation to study signal flow under different conditions of visual stimulation that include the border ownership paradigm.

Strengths: The paper's strengths are results of its use of multi-electrode probes to study border ownership in many neurons simultaneously across the cortical layers in V1. Also it provides new useful data about the dynamics of interaction of signals from the non-classical receptive field (NCRF) and the Classical receptive field (CRF).

Weaknesses:

The paper's weakness is that it does not challenge consensus beliefs about mechanisms. Also, the paper combines data about border ownership with data about the NCRF without making it clear how they are similar or different.

Critique:

The border ownership data on V1 offered in the paper replicate experimental results obtained by Zhou and von der Heydt (2000) and confirm the earlier results. The incremental addition is that the authors found border ownership in all cortical layers of V1, extending Zhou and von der Heydt's results that were only about layer 2/3 in V2 cortex. This is an interesting new result using the same stimuli but new measurement techniques.

The cross-correlation results show that the pattern of the cross correlogram (CCG) is influenced by the visual pattern being presented. However, in the initial submitted ms. the results were not analyzed mechanistically, and the interpretation was unclear. For instance, the authors show in Figure 3 (and in Figure S2) that the peak of the CCG can indicate layer 2/3 excites layer 4C when the visual stimulus is the border ownership test pattern, a large square 8 deg on a side. More than one reviewer asked, " how can layer 2/3 excite layer 4C"? . In the revised ms. the authors added a paragraph to the Discussion to respond to the reviewers about this point. The authors could provide an even better response to the reviewers by emphasizing that, consistently, layer 5/6 neurons lead neurons in layer 4, and for the CRF pattern and even more when the NCRF patterns are used.

The problems in understanding the CCG data are indirectly caused by the lack of a critical analysis of what is happening in the responses that reveal the border ownership signals, as in Fig.2. Let's put it bluntly--are border ownership signals excitatory or inhibitory? As the authors pointed out in their rebuttal, Zhang and von der Heydt (2010, JNS) did experiments to answer this question but I do not agree with the authors rebuttal letter about what Zhang and von der Heydt (2010) reported. If you examine Zhang and von der Heydt's Figure 6, you see that the major effect of stimulating border ownership neurons is suppression from the non-preferred side. That result is consistent with many papers on the NCRF (many cited by the authors) that indicate that it is mostly suppressive. That experimental fact about border ownership should be mentioned in the present paper.

What I should have pointed out in the first round, but didn't understand it then, is that there is a disconnect between the the border ownership laminar analysis (Figure 2) and the laminar correlations with CCGs (Figures 3-5) because the CCGs are not limited to border ownership neurons (or at least we are not told they were limited to them). So the CCG results are not mostly about border ownership--they are about the difference between signal flow in responses to small drifting Gabor patterns vs big flashed squares. Since only 21% of all recorded neurons were border ownership neurons, it is likely that most of the CCG statistics is based on neurons that do not show border ownership. Nevertheless, Figures 3 and 4 are very useful for the study of signal flow in the NCRF. It wasn't clear to me and I think the authors could make it clearer what those figures are about.

And I wonder if it might be possible to make a stronger link with border ownership by restricting the CCG analysis to pairs of neurons in which one neuron is a border ownership neuron. Are there enough data?

My critique of the CCG analysis applies to Figure 5 also. That figure shows a weak correlation of CCG asymmetry with Border Ownership Index. Perhaps a stronger correlation might be present if the population were restricted to the much smaller population of neuron pairs that had at least one border ownership neuron.

---

## [Author Response]

The following is the authors’ response to the original reviews.

**Reviewer #1 (Public review):**
Zhu and colleagues used high-density Neuropixel probes to perform laminar recordings in V1 while presenting either small stimuli that stimulated the classical receptive field (CRF) or large stimuli whose border straddled the RF to provide nonclassical RF (nCRF) stimulation. Their main question was to understand the relative contribution of feedforward (FF), feedback (FB), and horizontal circuits to border ownership (Bown), which they addressed by measuring crosscorrelation across layers. They found differences in cross-correlation between feedback/horizontal (FH) and input layers during CRF and nCRF stimulation.Although the data looks high quality and analyses look mostly fine, I had a lot of difficulty understanding the logic in many places. Examples of my concerns are written below.(1) What is the main question? The authors refer to nCRF stimulation emerging from either feedback from higher areas or horizontal connections from within the same area (e.g. lines 136 to 138 and again lines 223-232). I initially thought that the study would aim to distinguish between the two. However, the way the authors have clubbed the layers in 3D, the main question seems to be whether Bown is FF or FH (i.e., feedback and horizontal are clubbed). Is this correct? If so, I don't see the logic, since I can't imagine Bown to be purely FF. Thus, just showing differences between CRF stimulation (which is mainly expected to be FF) and nCRF stimulation is not surprising to me.

We thank the reviewer for their thoughtful comments. As explained in the discussion, we grouped cortical layers to reduce uncertainty in precisely assigning laminar boundaries and to increase statistical power. Consequently, this limits our ability to distinguish the relative contributions of feedback inputs, primarily targeting layers 1 and 6, and horizontal connections, mainly within layers 2/3 and 5. Nevertheless, previous findings, especially regarding the rapid emergence of B_own_ signals, suggest that feedback is more biologically plausible than horizontal-based mechanisms.

Importantly, the emergence of B_own_ signals in the primate brain should not be taken for granted. Direct physiological evidence that distinguishes feedforward from feedback/horizontal mechanisms has been lacking. While we agree it is unlikely that B_own_ is mediated solely by feedforward processing, we felt it was necessary to test this empirically, particularly using highresolution laminar recordings.

As discussed, feedforward models of B_own_ have been proposed (e.g., Super, Romeo, and Keil, 2010; Saki and Nishimura, 2006). These could, in theory, be supported by more general nCRF modulations arising through early feedforward inhibitions, such as those observed in the retinogeniculate pathway (e.g., Webb, Tinsley, Vincent and Derrington, 2005; Blitz and Regehr, 2005; Alitto and Usrey, 2008). However, most B_own_ models rely heavily on response latency, yet very few studies have recorded across layers or areas simultaneously to address this directly. Notably, recent findings in area V4 show that B_own_ signals emerge earlier in deep layers than in granular (input) layers, suggesting a non-feedforward origin (Franken and Reynolds, 2021).

Furthermore, although previous studies have shown that the nCRF can modulate firing rates and the timing of neuronal firing across layers, our findings go beyond these effects. We provide clear evidence that nCRF modulation also alters precise spike timing relationships and interlaminar coordination, and that the magnitude of nCRF modulation depends on these interlaminar interactions. This supports the idea that B_own_ , or more general nCRF modulation, involves more than local rate changes, reflecting layer-specific network dynamics consistent with feedback or lateral integration.

(2) Choice of layers for cross-correlation analysis: In the Introduction, and also in Figure 3C, it is mentioned that FF inputs arrive in 4C and 6, while FB/Horizontal inputs arrive at "superficial" and "deep", which I take as layer 2/3 and 5. So it is not clear to me why (i) layer 4A/B is chosen for analysis for Figure 3D (I would have thought layer 6 should have been chosen instead) and (ii) why Layers 5 and 6 are clubbed.

We thank the reviewer for raising this important point. The confusion likely stems from our use of the terms “superficial” and “deep” layers when describing the targets of feedback/horizontal inputs. To clarify, by “superficial” and “deep,” we specifically refer to layers 1–3 and layers 5–6, respectively, as illustrated in Figure 3C. Feedback and horizontal inputs relatively avoid entire layer 4, including both 4C and 4A/B.

We also emphasize that the classification of layers as feedforward or feedback/horizontal recipients is relative rather than absolute. For example, although layer 6 receives both feedforward and feedback/horizontal inputs, it contains a higher proportion of feedback/horizontal inputs compared to layers 4C and 4A/B.

We had addressed this rationale in the Discussion, but recognize it may not have been sufficiently emphasized. We have revised the main text accordingly to clarify this point for readers in the final manuscript version.

(3) Addressing the main question using cross-correlation analysis: I think the nice peaks observed in Figure 3B for some pairs show how spiking in one neuron affects the spiking in another one, with the delay in cross-correlation function arising from the conduction delay. This is shown nicely during CRF stimulation in Figure 3D between 4C -> 2/3, for example. However, the delay (positive or negative) is constrained by anatomical connectivity. For example, unless there are projections from 2/3 back to 4C which causes firing in a 2/3 layer neuron to cause a spike in a layer 4 neuron, we cannot expect to get a negative delay no matter what kind of stimulation (CRF versus nCRF) is used.

We thank the reviewer for the insightful comment. The observation that neurons within FH_i_ laminar compartments (layers 2/3, 5/6) can lead those in layer 4 (4C, 4A/B) during nCRF stimulation may indeed seem unexpected. However, several anatomical pathways could mediate the propagation of B_own_ signals from FH_i_ compartments to layer 4. We have revised the Discussion section in the final version of the manuscript to address this point explicitly.

In Macaque V1, projections from layers 2/3 to 4A/B have been documented (Blasdel et al., 1985; Callaway and Wiser, 1996), and neurons in 4A/B often extend apical dendrites into layers 2/3 (Lund, 1988; Yoshioka et al., 1994). Although direct projections from layers 2/3 to 4C are generally sparse (Callaway, 1998), a subset of neurons in the lower part of layer 3 can give off collateral axons to 4C (Lund and Yoshioka, 1991). Additionally, some 4C neurons extend dendrites into 4B, enabling potential dendritic integration of inputs from more superficial layers (Somogyi and Cowey, 1981; Mates and Lund, 1983; Yabuta and Callaway, 1998). Sparse connections from 2/3 to layer 4 have also been reported in cat V1 (Binzegger, Douglas and Martin, 2004). Moreover, layers 2/3 may influence 4C neurons disynaptically, without requiring dense monosynaptic connections.

Importantly, while CCGs can suggest possible circuit arrangements, functional connectivity may arise through mechanisms not fully captured by traditional anatomical tracing. Indeed, the apparent discrepancy between anatomical and functional data is not uncommon. For example, although 4B is known to receive anatomical input primarily from 4Cα, but not 4Cβ, photostimulation experiments have shown that 4B neurons can also be functionally driven by 4Cβ (Sawatari and Callaway, 1996). Our observation of functional inputs from layers 2/3 to layer 4 is also consistent with prior findings in rodent V1, where CCG analysis (e.g., Figure 7 in Senzai, Fernandez-Ruiz and Buzsaki, 2019) or photostimulation (Xu et al., 2016) revealed similar pathways.

Layers 5/6 provide dense projections to layers 4A/B (Lund, 1988; Callaway, 1998). In particular, layer 6 pyramidal neurons, especially the subset classified as Type 1 cells, project substantially to layer 4C (Wiser and Callaway, 1996; Fitzpatrick et al., 1985).

**Reviewer #2 (Public review):**
Summary:The authors present a study of how modulatory activity from outside the classical receptive field (cRF) differs from cRF stimulation. They study neural activity across the different layers of V1 in two anesthetized monkeys using Neuropixels probes. The monkeys are presented with drifting gratings and border-ownership tuning stimuli. They find that border-ownership tuning is organized into columns within V1, which is unexpected and exciting, and that the flow of activity from cellto-cell (as judged by cross-correlograms between single units) is influenced by the type of visual stimulus: border-ownership tuning stimuli vs. drifting-grating stimuli.Strengths:The questions addressed by the study are of high interest, and the use of Neuropixels probes yields extremely high numbers of single-units and cross-correlation histograms (CCHs) which makes the results robust. The study is well-described.Weaknesses:The weaknesses of the study are (a) the use of anesthetized animals, which raises questions about the nature of the modulatory signal being measured and the underlying logic of why a change in visual stimulus would produce a reversal in information flow through the cortical microcircuit and (b) the choice of visual stimuli, which do not uniquely isolate feedforward from feedback influences.(1) The modulation latency seems quite short in Figure 2C. Have the authors measured the latency of the effect in the manuscript and how it compares to the onset of the visually driven response? It would be surprising if the latency was much shorter than 70ms given previous measurements of BO and figure-ground modulation latency in V2 and V1. On the same note, it might be revealing to make laminar profiles of the modulation (i.e. preferred - non-preferred border orientation) as it develops over time. Does the modulation start in feedback recipient layers?(2) Can the authors show the average time course of the response elicited by preferred and nonpreferred border ownership stimuli across all significant neurons?

We thank the reviewer for the insightful comment—this is indeed an important and often overlooked point. As noted in the Discussion, B_own_ modulation differs from other forms of figure-ground modulation (e.g., Lamme et al., 1998) in that it can emerge very rapidly in early visual cortex—within ~10–35 ms after response onset (Zhou et al., 2000; Sugihara et al., 2011). This rapid emergence has been interpreted as evidence for the involvement of fast feedback inputs, which can propagate up to ten times faster than horizontal connections (Girard et al., 2001). Moreover, interlaminar interactions via monosynaptic or disynaptic connections can occur on very short timescales (a few milliseconds), further complicating efforts to disentangle feedback influences based solely on latency.

Thus, while the early onset of modulation in our data may appear surprising, it is consistent with prior B_own_ findings, and likely reflects a combination of fast feedback and rapid interlaminar processing. This makes it challenging to use conventional latency measurements to resolve laminar differences in B_own_ modulation. Latency comparisons are well known to be susceptible to confounds such as variability in response onset, luminance, contrast, stimulus size, and other sensory parameters.

Although we did not explicitly quantify the latency of B_own_ modulation in this manuscript, our cross-correlation analysis provides a more sensitive and temporally resolved measure of interlaminar information flow. We therefore focused on this approach rather than laminar modulation profiles, as it more directly addresses our primary research question.

(3) The logic of assuming that cRF stimulation should produce the opposite signal flow to borderownership tuning stimuli is worth discussing. I suspect the key difference between stimuli is that they used drifting gratings as the cRF stimulus, the movement of the stimulus continually refreshes the retinal image, leading to continuous feedforward dominance of the signals in V1. Had they used a static grating, the spiking during the sustained portion of the response might also show more influence of feedback/horizontal connections. Do the initial spikes fired in response to the borderownership tuning stimuli show the feedforward pattern of responses? The authors state that they did not look at cross-correlations during the initial response, but if they do, do they see the feedforward-dominated pattern? The jitter CCH analysis might suffice in correcting for the response transient.

We thank the reviewer for the insightful comment. As noted in the final Results section, our CRF and nCRF stimulation paradigms differ in respects beyond the presence or absence of nonclassical modulation, including stimulus properties within the CRF.

We agree with the reviewer’s speculation that drifting gratings may continually refresh the retinal image, promoting sustained feedforward dominance in V1, whereas static gratings might allow greater influence from feedback/horizontal inputs during the sustained response. Likewise, the initial response to the B_own_ stimulus could be dominated by feedforward activity before feedback/horizontal influences arrive.

This contrast was a central motivation for our experimental design: we deliberately used two stimulus conditions — drifting gratings to emphasize feedforward processing, and B_own_ stimuli, which are known to engage feedback modulation — to test whether these two conditions yield different patterns of interlaminar information flow. Our results confirm that they do. While we did not separately analyze the very initial spike period, our focus is on interlaminar information flow during the sustained response, which serves as the primary measure of feedback/horizontal engagement in this study.

Finally, beyond this direct comparison, we show in Figure 5 that under nCRF stimulation alone, the direction and strength of interlaminar information flow correlate with the magnitude of B_own_ modulation, further supporting the idea that our cross-correlation approach reveals functionally meaningful differences in cortical processing.

(4) The term "nCRF stimulation" is not appropriate because the CRF is stimulated by the light/dark edge.

We thank the reviewer for the comment. As noted in the Introduction, nCRF effects described in the literature invariably involve stimulation both inside and outside the CRF. Our use of the term “nCRF stimulation” refers to this experimental paradigm, rather than suggesting that the CRF itself is unstimulated. We hope this clarifies our use of the term.

**Reviewer #3 (Public review):**
Summary:The paper by Zhu et al is on an important topic in visual neuroscience, the emergence in the visual cortex of signals about figures and ground. This topic also goes by the name border ownership. The paper utilizes modern recording techniques very skillfully to extend what is known about border ownership. It offers new evidence about the prevalence of border ownership signals across different cortical layers in V1 cortex. Also, it uses pairwise cross-correlation to study signal flow under different conditions of visual stimulation that include the border ownership paradigm.Strengths:The paper's strengths are its use of multi-electrode probes to study border ownership in many neurons simultaneously across the cortical layers in V1, and its innovation of using crosscorrelation between cortical neurons -- when they are viewing border-ownership patterns or instead are viewing grating patterns restricted to the classical receptive field (CRF).Weaknesses:The paper's weaknesses are its largely incremental approach to the study of border ownership and the lack of a critical analysis of the cross-correlation data. The paper as it is now does not advance our understanding of border ownership; it mainly confirms prior work, and it does not challenge or revise consensus beliefs about mechanisms. However, it is possible that, in the rich dataset the authors have obtained, they do possess data that could be added to the paper to make it much stronger.Critique:The border ownership data on V1 offered in the paper replicates experimental results obtained by Zhou and von der Heydt (2000) and confirms the earlier results using the same analysis methods as Zhou. The incremental addition is that the authors found border ownership in all cortical layers extending Zhou's results that were only about layer 2/3.The cross-correlation results show that the pattern of the cross-correlogram (CCG) is influenced by the visual pattern being presented. However, the results are not analyzed mechanistically, and the interpretation is unclear. For instance, the authors show in Figure 3 (and in Figure S2) that the peak of the CCG can indicate layer 2/3 excites layer 4C when the visual stimulus is the border ownership test pattern, a large square 8 deg on a side. But how can layer 2/3 excite layer 4C? The authors do not raise or offer an answer to this question. Similar questions arise when considering the CCG of layer 4A/B with layer 2/3. What is the proposed pathway for layer 2/3 to excite 4A/B? Other similar questions arise for all the interlaminar CCG data that are presented. What known functional connections would account for the measured CCGs?

We thank the reviewer for raising this important point. As noted in our response to a previous comment, several anatomical pathways could mediate apparent functional inputs from layers 2/3 to 4C and 4A/B. In macaque V1, projections from layers 2/3 to 4A/B have been documented (Blasdel et al., 1985; Callaway and Wiser, 1996), and neurons in 4A/B often extend apical dendrites into layers 2/3 (Lund, 1988; Yoshioka et al., 1994). Although direct projections from layers 2/3 to 4C are generally sparse (Callaway, 1998), a subset of lower layer 3 neurons can give off collateral axons to 4C (Lund and Yoshioka, 1991). Some 4C neurons also extend dendrites into 4B, potentially allowing dendritic integration of inputs from more superficial layers (Somogyi and Cowey, 1981; Mates and Lund, 1983; Yabuta and Callaway, 1998). Sparse connections from 2/3 to layer 4 have also been reported in cat V1 (Binzegger et al., 2004).

Moreover, layers 2/3 may influence 4C neurons disynaptically, without requiring dense monosynaptic connections. While CCGs suggest possible circuit arrangements, functional connectivity may arise through mechanisms not fully captured by anatomical tracing, and apparent discrepancies between anatomical and functional data are not uncommon. For example, although 4B is known to receive anatomical input primarily from 4Cα, 4B neurons can also be functionally driven by 4Cβ using photostimulation (Sawatari and Callaway, 1996). Our observation of functional inputs from layers 2/3 to layer 4 is also consistent with prior findings in rodent V1, where CCG analysis (e.g., Figure 7 in Senzai, Fernandez-Ruiz and Buzsaki, 2019) or photostimulation (Xu et al., 2016) revealed similar pathways.

Layers 5/6 also provide dense projections to layers 4A/B (Lund, 1988; Callaway, 1998). In particular, layer 6 pyramidal neurons, especially the subset classified as Type 1 cells, project substantially to layer 4C (Wiser and Callaway, 1996; Fitzpatrick et al., 1985).

We have revised the Discussion section to explicitly address these points and clarify the potential anatomical and functional pathways underlying the measured interlaminar CCGs, highlighting how inputs from layers 2/3 and 5/6 to layer 4 can be mediated via both direct and indirect connections.

The problems in understanding the CCG data are indirectly caused by the lack of a critical analysis of what is happening in the responses that reveal the border ownership signals, as in Figure 2. Let's put it bluntly - are border ownership signals excitatory or inhibitory? The reason I raise this question is that the present authors insightfully place border ownership as examples of the action of the non-classical receptive field (nCRF) of cortical cells. Most previous work on the nCRF (many papers cited by the authors) reveal the nCRF to be inhibitory or suppressive. In order to know whether nCRF signals are excitatory or inhibitory, one needs a baseline response from the CRF, so that when you introduce nCRF signals you can tell whether the change with respect to the CRF is up or down. As far as I know, prior work on border ownership has not addressed this question, and the present paper doesn't either. This is where the rich dataset that the present authors possess might be used to establish a fundamental property of border ownership.Then we must go back to consider what the consequences of knowing the sign of the border ownership signal would mean for interpreting the CCG data. If the border ownership signals from extrastriate feedback or, alternatively, from horizontal intrinsic connections, are excitatory, they might provide a shared excitatory input to pairs of cells that would show up in the CCG as a peak at 0 delay. However, if the border ownership manuscript signals are inhibitory, they might work by exciting only inhibitory neurons in V1. This could have complicated consequences for the CCG.The interpretation of the CCG data in the present version of the m is unclear (see above). Perhaps a clearer interpretation could be developed once the authors know better what the border ownership signals are.

We thank the reviewer for raising this fundamental and thought-provoking question. As noted, B_own_ signals arise from nCRF, which has often been associated with suppressive effects. However, Zhang and von der Heydt (2010) provided important insight into this issue by systematically varying the placement of figure fragments outside the CRF while keeping an edge centered within the CRF. They found that contextual fragments on the preferred side of B_own_ produce facilitation, while those on the non-preferred side produce suppression. Thus, the nCRF contribution to B_own_ reflects both excitatory and inhibitory modulation, depending on the spatial configuration of the figure.

These effects were well explained by their model in which feedback from grouping cells in higher areas selectively enhances or suppresses V1/V2 neuron responses, depending on their B_own_ preference. In this framework, the B_own_ signal itself is not inherently excitatory or inhibitory; rather, it results from the net effect of feedback, which can be either facilitative or suppressive. Importantly, it is the input that is modulated — not that the receiving neurons are necessarily inhibitory themselves.

In the current study, our analysis focused on CCGs showing excessive coincident spiking, i.e., positive peaks, which are typically interpreted as evidence for shared excitatory input or excitatory connections. Due to the limited number of connections, we did not analyze inhibitory interactions, such as anti-correlations or delayed suppression in the CCGs, which would be expected if the reference neuron were inhibitory. Therefore, the CCGs we report here likely reflect the excitatory component of the B_own_ signal, and possibly its upstream drive via feedback. While a full separation of excitatory and inhibitory components remains an important goal for future work, our data suggest that B_own_ modulation is at least partially mediated through excitatory feedback input.

My critique of the CCG analysis applies to Figure 5 also. I cannot comprehend the point of showing a very weak correlation of CCG asymmetry with Border Ownership Index, especially when what CCG asymmetry means is unclear mechanistically. Figure 5 does not make the paper stronger in my opinion.

We thank the reviewer for this comment. As described in the Results section for Figure 5, the observation that interlaminar information flow correlates with B_own_ modulation is important because it demonstrates that these flow patterns are specifically related to the magnitude of B_own_ signals, independent of the comparisons between CRF and nCRF stimulation.

In Figure 3, the authors show two CCGs that involve 4C--4C pairs. It would be nice to know more about such pairs. If there are any 6--6 pairs, what they look like also would be interesting. The authors also in Figure 3 show CCG's of two 4C--4A/B pairs and it would be quite interesting to know how such CCGs behave when CRF and nCRF stimuli are compared. In other words, the authors have shown us they have many data but have chosen not to analyze them further or to explain why they chose not to analyze them. It might help the paper if the authors would present all the CCG types they have. This suggestion would be helpful when the authors know more about the sign of border ownership signals, as discussed at length above.

We thank the reviewer for the insightful comment. The rationale for selecting specific laminar pairs is described in the Results section after Figure 3C and further discussed in the Discussion. In brief, we focused on CCGs computed from pairs in which one neuron resided in laminar compartments receiving feedback/horizontal inputs (layers 2/3 and 5/6) and the other within compartments relatively devoid of these inputs (layers 4C and 4A/B).

To mitigate uncertainty in defining exact laminar boundaries and to maximize statistical power, we combined some anatomical layers into distinct laminar compartments. This approach allowed us to compare the relative spike timing between neuronal pairs during CRF and nCRF stimulation. If feedback/horizontal inputs contribute more during nCRF than CRF stimulation, we expect this to be reflected in the lead-lag relationships of the CCGs. While other pairs (e.g., 5/6–5/6 or 4C– 4A/B) could in principle be analyzed, the hypothesized patterns for these pairs are less clear, and thus they were not the focus of our study. Nonetheless, these additional pairs represent interesting directions for future work.